# ToolQA: A Dataset for LLM Question Answering with External Tools

**Yuchen Zhuang***, **Yue Yu***, **Kuan Wang***, **Haotian Sun, Chao Zhang**
College of Computing, Georgia Institute of Technology, Atlanta GA
{yczhuang, yueyu, kuanwang, haotian.sun, chaozhang}@gatech.edu

## Abstract

Large Language Models (LLMs) have demonstrated impressive performance in various NLP tasks, but they still suffer from challenges such as hallucination and weak numerical reasoning. To overcome these challenges, external tools can be used to enhance LLMs' question-answering abilities. However, current evaluation methods do not distinguish between questions that can be answered using LLMs' internal knowledge and those that require external information through tool use. To address this issue, we introduce a new dataset called ToolQA, which is designed to faithfully evaluate LLMs' ability to use external tools for question answering. Our development of ToolQA involved a scalable, automated process for dataset curation, along with 13 specialized tools designed for interaction with external knowledge in order to answer questions. Importantly, we strive to minimize the overlap between our benchmark data and LLMs' pre-training data, enabling a more precise evaluation of LLMs' tool-use reasoning abilities. We conducted an in-depth diagnosis of existing tool-use LLMs to highlight their strengths, weaknesses, and potential improvements. Our findings set a new benchmark for evaluating LLMs and suggest new directions for future advancements. Our data and code are freely available for the broader scientific community on GitHub [2].

## 1 Introduction

Large Language Models (LLMs) have demonstrated superior performance in a myriad of NLP tasks [4, 9, 43, 42, 55, 61]. These models have captured vast amounts of knowledge from enormous and diverse corpora during pre-training. After instruction fine-tuning [10, 44, 2], they have demonstrated impressive capabilities in information-seeking question answering [65, 26]. Despite their remarkable performance, LLMs face several challenges. For example, they are susceptible to hallucinations—generating plausible yet ungrounded information—which can mislead users and affect content integrity [66, 19, 5]. Additionally, they exhibit weaknesses in numerical reasoning, an essential skill in numerous real-life applications [14, 36, 41, 28, 51, 13]. These limitations highlight the need for techniques that can enhance LLMs' question-answering abilities.

Recent research has shown that these issues can be mitigated by augmenting LLMs with *external tools*, such as retrieval augmentation [58, 17], math tools [56, 76, 32], and code interpreters [13, 63]. For example, a Wolfram math plugin can enhance numerical reasoning [68], and a verified database can mitigate hallucinations by providing up-to-date fact-checked knowledge [49]. However, existing evaluation methodologies struggle to distinguish whether the model is simply recalling pre-trained information or truly utilizing external tools for problem-solving [37]. This challenge arises, in part, because the external data used for evaluation may have already been exposed to LLMs during the pre-training phase [53]. This exposure can lead to a biased evaluation of LLMs' tool-use abilities, as the models could just use their ingrained knowledge and their reasoning abilities, bypassing the use of external tools. As a result, these evaluations cannot accurately reflect the true competency of the

---

*These authors contributed equally to this work.
[2] https://github.com/night-chen/ToolQA

37th Conference on Neural Information Processing Systems (NeurIPS 2023) Track on Datasets and Benchmarks.

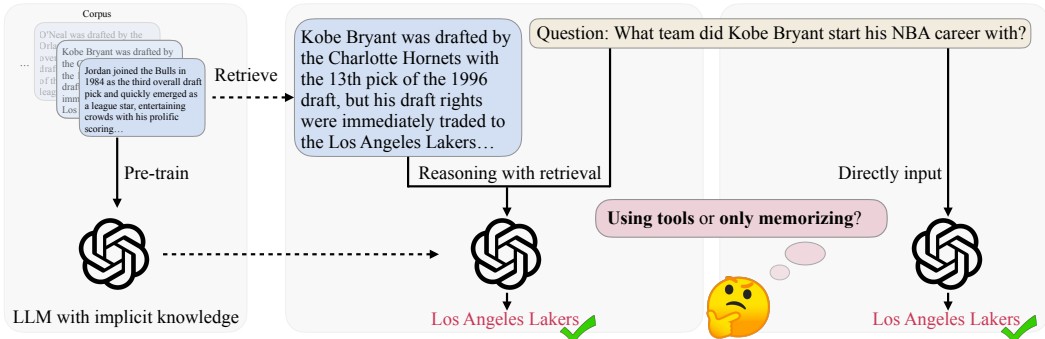

Figure 1: Pre-trained on vast range of corpus, LLMs possess extensive knowledge, which may overlap with evaluation data. This overlap poses a significant challenge to current evaluation methods, as it becomes difficult to discern whether the model is merely recalling pre-trained information or genuinely employing external tools for problem-solving.

models. We need a fair and explicit way to check if LLMs are really good at problem-solving with tools or if they are just using their memorized information.

To fill this gap, we introduce ToolQA, a question answering (QA) benchmark to evaluate LLMs' ability in using external tools for answering questions. ToolQA comprises data from 8 domains and defines 13 types of tools to acquire information from external reference corpora. Each instance in ToolQA consists of a question, an answer, reference corpora, and a list of available tools. ToolQA is unique in that all its questions can be answered only by using appropriate tools to obtain information from the reference corpus. This minimizes the possibility of LLMs answering questions by merely recalling their internal knowledge, and allows for faithfully evaluating LLMs' abilities in using tools.

ToolQA is curated with an automated three-phase process: (1) The first phase, *Reference Data Collection*, involves gathering various types of public corpora including text, tables, and graphs from different domains. These corpora have no overlap with the LLM pre-training data and will serve as reference corpora for tool-based question answering. (2) The second phase is *Human-guided Question Generation with LLMs*. In this phase, we generate questions that can only be answered by using tools over the reference corpora. Our approach is a *template-based* question generation process, which includes human-guided template generation, template validation, and question instantiation with tool attributes. (3) The third phase is *Programmatic Answer Generation*. This phase produces accurate answers for the generated questions. To ensure answer correctness, we implement operators corresponding to the tools and obtain answers from the reference corpora programmatically. Our three-phase procedure ensures that we generate questions that can only be answered using external knowledge, along with their precise answers. Additionally, the process is highly efficient and requires minimal human labeling efforts.

We conducted experiments using both standard LLMs and tool-augmented LLMs to answer questions in ToolQA. Our findings indicate that ChatGPT and Chain-of-thoughts prompting [65], which rely solely on their internal knowledge, have low success rates of approximately 5% for easy questions and 2% for hard questions. In contrast, tool-augmented LLMs such as Chameleon [32] and ReAct [76] perform better by leveraging external tools. For easy questions, the best performance achieved by tool-augmented LLMs is 43.15%, while for hard questions, the best performance drops to 8.2%. Our results and error analysis demonstrate that ToolQA is a challenging benchmark for existing tool-augmented LLM methods, especially for its hard questions that require more complex reasoning about tool composition.

## 2   Related Work

### 2.1   Knowledge-Augmented LLMs

Several prior works aim to enhance LLMs with explicit external knowledge. Specifically, one line of research focus on *retrieval-augmented language models* [58, 3, 17, 27, 30, 80, 34, 72], where they use sparse [54] or dense retrieval [22, 16] to extract relevant knowledge from the corpus. These works mainly focus on leveraging free text, without considering multiple types of tools for task solving. On the other hand, Program-of-Thought [6], PAL [13], MathPrompt [15], and

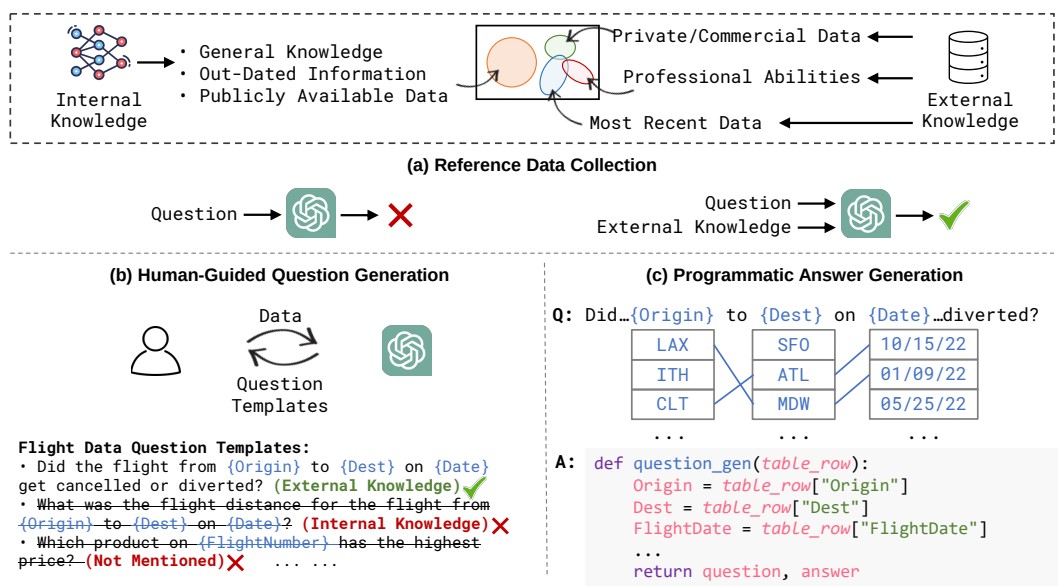

Figure 2: ToolQA, aiming to faithfully evaluate LLMs' abilities to use external tools, curates data through three phases: (a) Reference Data Collection; (b) Human-Guided Question Generation; and (c) Programmatic Answer Generation.

Code4Struct [63] apply code-based tools to enhance LLMs' abilities in question answering with a focus on tabular and math-related tasks. Several additional works [56, 32, 57] expand the scope of tool utilization by incorporating different types of basic tools (*e.g.* calculator, calendar, machine translation) to solve complex reasoning tasks. To synergize different functional tools together for problem-solving, LLMs must have advanced planning and memory capabilities. In terms of planning, current methods either enable LLMs to autonomously break down complex tasks into intermediate reasoning steps [45, 24, 47, 50, 65, 75, 31, 76], or encourage LLMs to self-reflect the previous decisions with environmental feedback [39, 76, 35, 8]. Memory capabilities, on the other hand, provide LLMs with opportunities to learn and adapt based on past experiences, whether successes or failures [62]. In addition, several works have extended this line of learning paradigm to other modalities [73, 69] and other domains [20]. Concurrent to our work, there are also several studies [70, 37] that investigate the parametric and nonparametric knowledge from LLMs. A detailed comparison between existing tool-use LLMs can be found in Appendix A.

## 2.2 Benchmarks on Tool-Augmented LLMs

Earlier tool-augmented LLMs primarily assess single tool usage based on downstream task performance across existing benchmarks. For example, there are works that study how text retrievers augment LLMs' performance on open-domain question-answering [21, 74], fact-checking [60], and timely information benchmarks [7, 23, 78, 12]. Besides, the mathematical reasoning abilities of external calculators and Python interpreters are evaluated using computation-intensive QA datasets [11, 33]. However, these evaluation benchmarks may not faithfully reflect the extent to which models leverage external tools, as some questions could still be correctly answered solely using the internal knowledge of the LLMs. ToolQA attempts to mitigate these issues by selecting data from out-of-scope sources that have not been memorized by LLMs. Concurrent with our work, there are several recent benchmarks for evaluating LLMs' ability in using multiple tools for solving challenging tasks, including API-Bank [29], APIBench [48], and ToolBench [52, 71]. They mainly focus on constructing high-quality tool chains for LLM fine-tuning and evaluating API call trace accuracy against a fixed ground truth trace. In contrast, ToolQA is unique in that it focuses on the open-ended use of tools for question-answering, rather than benchmarking the intermediate process of tool use. Specifically, ToolQA creates tool-based question-answer pairs and assesses whether LLMs can arrive at the correct answer, regardless of the tool chains used.

# 3 ToolQA Dataset

## 3.1 Dataset Details

We curate the ToolQA benchmark to evaluate LLMs' capability in leveraging external tools for question answering. ToolQA consists of data from 8 distinct domains, each instance being a tuple — (*question*, *answer*, *reference corpora*, and *tools*). The *reference corpora* are external knowledge sources that can be queried, which can be a text corpus, a tabular database, or a graph. To enable obtaining information from the reference corpora, we have developed 13 tools for text retrieval, database operations, code interpretation, mathematical computations, and more. The questions are designed to simulate real-world information-seeking inquiries. However, they cannot be answered directly with LLMs' internal knowledge, but instead require LLMs to obtain information from the reference corpora via tool use. Table 1 shows the detailed statistics of ToolQA.

To reduce human efforts in generating faithful question-answer pairs to evaluate LLMs' tool-use capabilities, we propose an automatic three-phase process (Figure 2): (1) We first select data from public sources that are unmemorized by LLMs during *Reference Data Collection*; (2) We adopt *Human-Guided Question Generation* to steer LLMs to generate valid questions according to pre-defined templates; (3) We produce accurate answers for the generated questions with *Programmatic Answer Generation*. We detail the three-phase generation process in the following.

## 3.2 Reference Data and Tools

To evaluate LLMs' ability in using external tools for question answering, it is crucial to ensure that they cannot directly answer the questions with their internal knowledge. To this end, we collect reference corpora that meet the following criteria (Figure 2(a)): 1) The reference corpora should ideally not overlap with the LLM's pre-training data; 2) The reference corpora should contain context-sensitive facts for generating questions that cannot be directly answered solely based on LLMs' internal knowledge and reasoning abilities; 3) LLMs should be able to obtain all the necessary information from the reference corpora to correctly answer the questions.

Based on these criteria, we define 6 contextual dimensions: *temporal*, *spatial*, *social*, *scientific*, *mathematical*, and *personal*. We collect reference corpora that can yield *context-specific* questions along one or more of the 6 dimensions. Specifically: 1) Along the *temporal* dimension, we collect the `Flights` and `Coffee` corpora, which contain the latest information that is out of the temporal scope of the LLM's pre-training data. 2) Along the *spatial* dimension, we collect `Yelp` and `Airbnb`, which are two non-text corpora that can yield questions with spatial contexts. 3) Along the *mathematical* dimension, we collect the questions from `GSM8K` that ChatGPT cannot answer correctly with its own mathematical reasoning ability; 4) `SciREX` emphasizes detailed model performances from the *scientific* domain [18], where GPT family models can easily hallucinate [42]. 5) To incorporate *personal* data and avoid privacy issues, we synthesize the personal `Agenda` corpus with ChatGPT with virtual names and events. 6) In addition, we also select data from the most recent DBLP database and create graphs between authors and papers, where *social* relational knowledge cannot be understood by LLMs currently. Further details can be found in Appendix B.

To obtain information from these reference corpora, we design 13 tools that are available to the LLMs (Table 2). These tools are designed as follows:

Table 1: Dataset Statistics of ToolQA.

| Context | Topic | External Knowledge | | Easy | | Hard | |
|---|---|---|---|---|---|---|---|
| | | Format | Size | # Templates | # Questions | # Templates | # Questions |
| Temporal | Flight | Tabular Database | 4078318 | 10 | 100 | 10 | 100 |
| | Coffee | Tabular Database | 5746 | 8 | 100 | 13 | 130 |
| Spatial | Yelp | Tabular Database | 150346 | 11 | 100 | 10 | 100 |
| | Airbnb | Tabular Database | 102599 | 10 | 100 | 10 | 100 |
| Mathematical | GSM8K | Professional Ability | - | - | 100 | - | - |
| Social | DBLP | Graph | 553320 | 10 | 100 | 10 | 100 |
| Scientific | SciREX | Pure-Text Corpus | 438 | 1 | 100 | 4 | 100 |
| Personal | Agenda | Pure-Text Corpus | 10000 | 5 | 100 | 5 | 100 |
| **SUM** | - | - | - | **55** | **800** | **62** | **730** |

Table 2: Different tools in ToolQA.

| Tool Types | # Tools | Tools |
|---|---|---|
| Text Tools | 2 | Agenda Retriever, SciREX Retriever |
| Database Tools | 3 | Database Loader, Data Filter, Get Value |
| Math Tools | 1 | WolframAlpha Calculator |
| Graph Tools | 4 | Graph Loader, Neighbour Checker, Node Checker, Edge Checker |
| Code Tools | 2 | Python Interpreter, SQL Interpreter |
| System Tools | 1 | Finish |

- **Text:** *AgendaRetriever* and *SciREXRetreiver* are text retrieval tools. They can retrieve relevant information to a given query from the (synthesized) personal agenda corpus and scientific corpus.
- **Database:** *Database Loader* loads data from the local tabular Database. *Data Filter* can filter the database according to a set of conditions, each of which is composed of a column name, a relation, and a pre-determined value (*e.g.*, "`Date=2022-10-15`"). *Get Value* returns all the values under a certain column in the database.
- **Math:** *Calculator* is a mathematical tool that treats the input string as a formula and calculates the corresponding result. We use the WolframAlpha API portal as the calculator [3], which can perform both simple computations (*e.g.*, add, subtraction, multiplication) and complicated operations (*e.g.*, averaging, finding maximum values).
- **Graph:** *Graph Loader* loads the graph from local files for future operations. *Neighbour Checker* lists all the neighbors of the query node in the graph. *Node Checker* and *Edge Checker* return the detailed attribute information of the query node and edge, respectively.
- **Code:** The *SQL Interpreter* and the *Python Interpreter* are responsible for interpreting and executing SQL commands and Python code, respectively. They can receive and transform data from other tools, serving as bridges between different tools and the LLM.
- **System:** *Finish* parses the feedback from execution and returns the answer to finish the task.

## 3.3 Human-Guided Question Generation

The question generation phase aims to generate questions that can be answered by using the available tools over the reference corpora. There are two straightforward strategies to generate questions: 1) letting human experts come up with questions about reference corpora, or 2) relying solely on LLMs to generate questions about the reference corpora. However, both strategies have their drawbacks. While human experts can produce high-quality questions, the entire process is labor-intensive, time-consuming, and hard to scale. Depending solely on LLMs may generate unanswerable questions or hallucinate information that does not exist in the reference data. Besides, some of the LLM-generated questions are too easy and can be directly answered with only LLMs' internal knowledge.

To address these challenges, we propose a human-guided LLM generation approach that uses question templates to bridge human guidance and automatic LLM generation [67, 79]. We first ask ChatGPT to generate *candidate question templates* from reference data, using prompts such as "*Generate diverse and challenging template questions that users may have based on the given information.*". To obtain diverse questions, we generate around 50 template questions for each external data source. We then perform manual validation to select the templates that cannot be answered with LLMs' internal knowledge but become answerable with the reference corpora. We go through all question templates and eliminate those that meet either of the following conditions: (1) Template questions that vanilla ChatGPT can answer based on its internal knowledge with a success rate of over $50\%$ (e.g., "What is the distance between LAX and SFO?", where distance information can be memorized by ChatGPT); (2) Template questions posing queries about information not present in the external data (e.g., "What is the average price from LAX to SFO?", where price information is missing from the flight data). After examining all the templates, we selected 117 most representative and diverse question templates for the entire ToolQA dataset.

After the high-quality question templates are manually selected, we sample values from the reference data to automatically fill into the templates to generate concrete questions. For example, given the template "*Did the flight from {Origin} to {Dest} on {Date} get canceled or diverted?*", we can sample the values "`LAX`", "`MDW`", "`01/09/22`" from the reference Flight tabular data and fill into the template to form a question: "*Did the flight from `LAX` to `MDW` on `01/09/22` get canceled or diverted?*"

---

[3] https://products.wolframalpha.com/api

Depending on the difficulty of the questions, we classify them into two classes — easy and hard. Easy questions primarily focus on extracting a single piece of information from external knowledge, thus requiring fewer tools to involve in the solution. Conversely, hard questions require complex operations (*e.g.*, average) and reasoning (*e.g.*, comparison) over multiple information pieces drawn from the reference corpora, requiring more tools and complex reasoning among them. We provide a comprehensive list of both easy and hard question templates in Appendix C and D.

### 3.4 Programmatic Answer Generation

Our final step is to create accurate answers for the generated questions. To guarantee the validity of these responses, we implement 1) operators, which are functions corresponding to the predefined tools; and 2) tool chains, which are schemas for composing different operators for different question templates. For each question, as we know the true arguments filled into the question template, we can run the tool chains with the corresponding arguments to programmatically extract answers from the reference data. This process enables automatic generation correct answers to questions, even for those questions that involve multi-step reasoning. Figure 2(c) demonstrates this generation process. When answering a generated question with sampled values "*Did the flight from* LAX *to* MDW *on* 01/09/22 *get canceled or diverted?*", we write Python codes to implement the operators over the reference data, including database loader, data filter, and get-value function. Then, the programmatic pipeline runs a tool chain of these operators to automatically generate the correct answer (details in Appendix E).

## 4 Experiments

### 4.1 Baselines

We evaluate the performance of the following methods on ToolQA, covering both standard LLMs and tool-augmented LLMs: (1) **LLaMA-2** [61] and **Falcon** [1] are state-of-the-art open-sourced large language models. We directly feed the questions into two versions (13B and 70B) of LLaMA-2 and Falcon (40B) to obtain the predictions; (2) **ChatGPT** [43]: We directly feed the questions into OpenAI's ChatGPT model (`gpt-3.5-turbo`) and obtain its response as the final answer. (3) **CoT** [65, 26]: We use chain-of-thoughts prompting for ChatGPT, adding the prompt "Let's think step by step:" after the question to leverage LLMs' reasoning ability for question answering. (4) **Chameleon** [32] is a recent method that uses LLMs as a controller to use multiple tools for solving subtasks and has shown promising results in reasoning and QA tasks. When running Chameleon on ToolQA, we set the tool pool to our defined tools in § 3.1. (5) **ReAct** [76] integrates reasoning with tool use by prompting LLMs to generate interleaved verbal reasoning traces and tool calls. This integration has been shown effective in enhancing LLMs' problem-solving capabilities. We instantiate two versions of ReAct using `gpt-3.5-turbo` and `text-davinci-003`.

Different from the existing works that mainly provide task-level few-shot exemplars, we provide tool-level demonstrations. We used 8 demonstrations about how to use tools for QA, ensuring that each tool in the pool is covered at least once by the demonstrations. Such tool-level demonstrations provide a concise tutorial to the LLMs for tool use, covering all tool uses with the LLM context limit. Details about the demonstrations and our prompts are included in Appendix F. To assess the performance of methods on the ToolQA benchmark, we normalize both the ground-truth answers and the model predictions to ensure uniformity in format. Success rates are then computed based on the exact match between these normalized answers. We use a series of rules for normalization: (1) We normalize different time string formats, (*e.g.*, converting "18:06" and "1806.0" to "1806".); (2) For price-related questions, we normalize the units by removing price units (*e.g.*, USD, $); (3) We remove all the punctuations from both the model predictions and ground-truth answers; (4) We normalize the article usage (*e.g.*, a, an, the) via removing all articles from both the model predictions and ground-truth answers; (5) We normalize the white spaces by trimming multiple spaces into single space. As most of the predictions and answers are numerical values or entities, these normalization rules address most of the false negative cases during matching. We evaluate the model's ability against the generated question-answer pairs in an open-ended manner, focusing on whether the model can arrive at the correct answer, regardless of the used tool chains.

### 4.2 Results

**Internal Knowledge vs. External Knowledge.** From the results in Tables 3 and 4, the vanilla open-sourced LLMs and ChatGPT underperform their tool-augmented counterparts on both easy and hard questions. This is expected, as vanilla LLMs lack access to external information for question

Table 3: Success rates on easy questions.

| LLM Category | Models | Flight | Coffee | Agenda | Yelp | DBLP | SciREX | GSM8K | Airbnb | Average |
|---|---|---|---|---|---|---|---|---|---|---|
| Open-Sourced LLMs | LLaMA-2 (13B) | 0.0 | 2.0 | 0.0 | 5.0 | 1.0 | 0.0 | 9.0 | 1.0 | 2.3 |
| | Falcon (40B) | 1.0 | 1.0 | 2.0 | 8.0 | 1.0 | 0.0 | 8.0 | 5.0 | 3.3 |
| | LLaMA-2 (70B) | 2.0 | 6.0 | 5.0 | 15.0 | 0.0 | 0.0 | 9.0 | 4.0 | 5.1 |
| Closed-Sourced LLMs | ChatGPT | 2.0 | 0.0 | 0.0 | 15.0 | 0.0 | 2.0 | 26.0 | 0.0 | 5.6 |
| | CoT | 1.0 | 1.0 | 0.0 | 9.0 | 0.0 | 0.0 | 30.0 | 0.0 | 5.1 |
| Tool-Augmented LLMs | Chameleon | 30.0 | 9.0 | 4.0 | 8.0 | 3.0 | 0.0 | 27.0 | 4.0 | 10.6 |
| | ReAct (GPT-3) | **61.0** | **90.0** | **29.0** | **77.0** | **28.0** | **3.0** | **32.0** | 25.0 | **43.1** |
| | ReAct (GPT-3.5) | 48.0 | 81.0 | 24.0 | 64.0 | 23.0 | 2.0 | 23.0 | **29.0** | 36.8 |

Table 4: Success rate on hard questions.

| LLM Category | Models | Flight | Coffee | Agenda | Yelp | Airbnb | DBLP | SciREX | Average |
|---|---|---|---|---|---|---|---|---|---|
| Open-Sourced LLMs | LLaMA-2 (13B) | 1.0 | 0.0 | 0.0 | 4.0 | 1.0 | 5.0 | 1.0 | 1.7 |
| | Falcon (40B) | 1.0 | 0.0 | 0.0 | 4.0 | 1.0 | 6.0 | 1.0 | 1.9 |
| | LLaMA-2 (70B) | 1.0 | 0.0 | 0.0 | 4.0 | 1.0 | 4.0 | 3.0 | 1.9 |
| Closed-Sourced LLMs | ChatGPT | 2.0 | 2.3 | 1.0 | 0.0 | 2.0 | 4.0 | 3.0 | 2.0 |
| | CoT | 0.0 | 0.8 | 0.0 | 1.0 | 0.0 | 3.0 | 5.0 | 1.4 |
| Tool-Augmented LLMs | Chameleon | 3.0 | 2.3 | 0.0 | 0.0 | 0.0 | 8.0 | 0.0 | 1.9 |
| | ReAct (GPT-3) | 3.0 | 10.8 | 0.0 | 3.0 | 0.0 | **19.0** | 0.0 | 5.1 |
| | ReAct (GPT-3.5) | **5.0** | **17.7** | **7.0** | **8.0** | **7.0** | 5.0 | **8.0** | **8.2** |

answering. Additionally, the vanilla LLMs show near-zero performance on different tasks, indicating that there is little overlap between the benchmark data and the LLMs' internal knowledge.

**Comparing Different Tool-Use LLMs.** Tables 3 and 4 show the results of different methods on the easy and hard questions. ChatGPT and CoT achieve very poor success rates ($< 10$) on both easy and hard questions across different tasks. This is expected as the questions in ToolQA cannot be answered solely based on LLMs' internal knowledge and reasoning. Chameleon achieves slightly better performance, with 10.6% and 1.9% success rates on easy and hard questions, respectively. This is because Chameleon incorporates tool descriptions and integrates human-induced orderings of these tools in its context, enabling it to comprehend and compose different tools for QA. However, Chameleon cannot take feedback from the execution trace, thus often suffering from infeasible actions or omitted arguments in its generated plans. ReAct is the best-performing model. It can use observations in the execution trace to generate its next action, allowing it to iteratively refine its tool use chain and obtain better success rates.

**Easy vs. Hard Questions.** Comparing Tables 3 and 4, we observe that all the baselines perform much worse on hard questions. The best method achieves an average success rate of $43.13\%$ on easy questions, while that number drops to $8.24\%$ on hard questions. As mentioned in § 3, the hard questions in ToolQA require more tool calls and more complicated compositions. Current tool-augmented LLMs struggle with answering such hard questions, which requires further development of techniques to improve their ability to reason about the task and generate plans for tool use.

**GPT-3 vs. GPT3.5.** [4] Comparing the different versions of ReAct, we observe that the ReAct (GPT-3) outperforms ReAct (GPT-3.5) on easy questions, yet it shows inferior performance on hard questions. Our hypothesis is that for easy questions, it is more important to learn and follow the format of the tool calls in the context, which GPT-3 is stronger at. For hard questions, the better reasoning and code understanding abilities of GPT-3.5 enables it to come up with "innovative" solutions that never appear in the context, leading to higher success rates. An example can be referred to in § 5.3.

## 5 Result Analysis and Discussion

We analyze the drawbacks and possible improvements of existing tool-augmented LLMs, taking the best-performed ReAct (GPT-3.5) model on the hard questions of ToolQA as an example.

### 5.1 Main Error Type I: Argument Errors

By performing comprehensive error analysis, we found that the most common error type when asking LLMs to use tools for QA is argument error — LLMs calling the tools with wrong arguments.

---

[4]GPT-4 was not included in the evaluation as we have no access to its API.

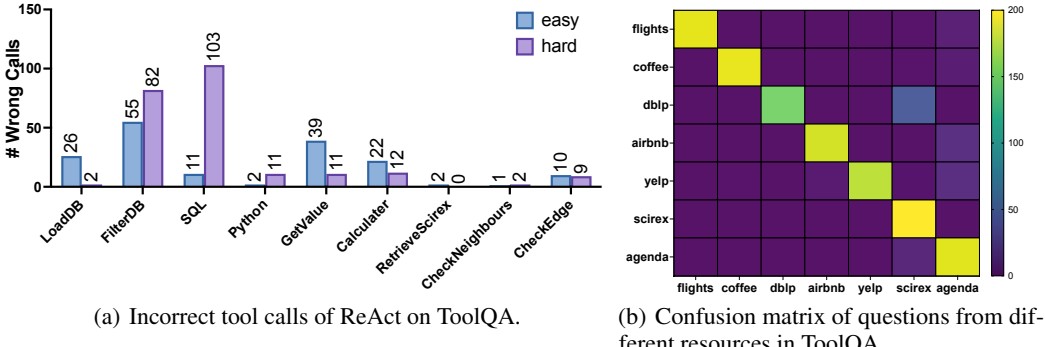

(a) Incorrect tool calls of ReAct on ToolQA.

(b) Confusion matrix of questions from different resources in ToolQA.

Figure 3: Analysis of incorrect tool calls and incorrect data sources made by ReAct on ToolQA.

For ReAct, this error type makes $44.56\%$ and $48.23\%$ out of the $377$ and $436$ error cases on easy and hard questions respectively, as shown in Figure 3(a). Interestingly, ReAct shows different argument error patterns on easy and hard questions. On easy questions, it tends to make more mistakes on database-related tools. For example, the model commits a total of 120 errors when calling `LoadDB`, `FilterDB`, and `GetValue` tools for easy questions, while this number reduces to 95 for hard questions. On the other hand, when dealing with code-related tools (*e.g.*, `SQLInterpreter` and `PythonInterpreter`), ReAct makes nearly 10x more errors for hard questions than for easy ones. This phenomenon is likely because the solution logic for hard questions is often more complex and cannot be fully inferred from the context alone. Consequently, the LLMs tend to rely on their understanding of code and programming concepts to tackle these intricate questions. In contrast, for easy questions, the LLMs tend to follow the patterns provided in the context, attempting to combine different database operations to arrive at a solution.

## 5.2 Main Error Type II: Incorrect Data Source

We have conducted an investigation into the data sources preferred by LLMs when answering questions. We found that LLMs also have difficulties in identifying the proper reference corpora answer the questions. This behavior is graphically represented as a confusion matrix in Figure 3(b). Upon examining the figure, it is apparent that for target reference corpora like Flight, Coffee, Airbnb, and Yelp that contain temporal information, LLMs are more likely to query the Agenda corpus for answering questions. Similarly, given that the SciREX knowledge corpora and DBLP graph are both in the scientific domain, LLMs tend to be confused about which source to query when answering scientific questions.

## 5.3 Main Error Type III: Innovation and Hallucination

For in-context tool-augmented LLMs, it is typical to include descriptions and use-case examples of each tool in the prompt. However, as the problem complexity increases with the number of tools, it becomes challenging to encompass all possible instances of compositional tool use as few-shot exemplars. Consequently, it is vital for LLMs to uncover logical relationships among different tools, which have never been encompassed in the human-provided exemplars, to solve challenging tasks — a process we refer to as "innovation." However, these innovative behaviors are a double-edged sword as they are often accompanied by hallucinations. Figure 4 illustrates this phenomenon with a case study, where LLMs answer hard questions with reference `Coffee` data. Given the context length constraint, the few-shot exemplar only showcases the basic usage of database operations and the SQL interpreter. For the hard question in Figure 4(a), ReAct (GPT-3) strictly follows the operations displayed in the context, leading to failure. On the contrary, ReAct (GPT-3.5) innovatively identifies the SQL interpreter as a possible alternative to database operations, especially when the latter fails repeatedly. However, such innovations can oftentimes lead to hallucinations. As shown in Figure 4(b), when answering another hard question from the Coffee data, ReAct (GPT-3.5) opts to hallucinate certain observations (highlighted in yellow) that are non-existent in the feedback from tool execution.

## 5.4 Other Error Types

We manually go through and count all the errors made by the ReAct (GPT-3.5) model and show the errors on both easy and hard questions in Figure 5. In addition to the aforementioned 3 main error types, there are 4 error types that frequently occur:

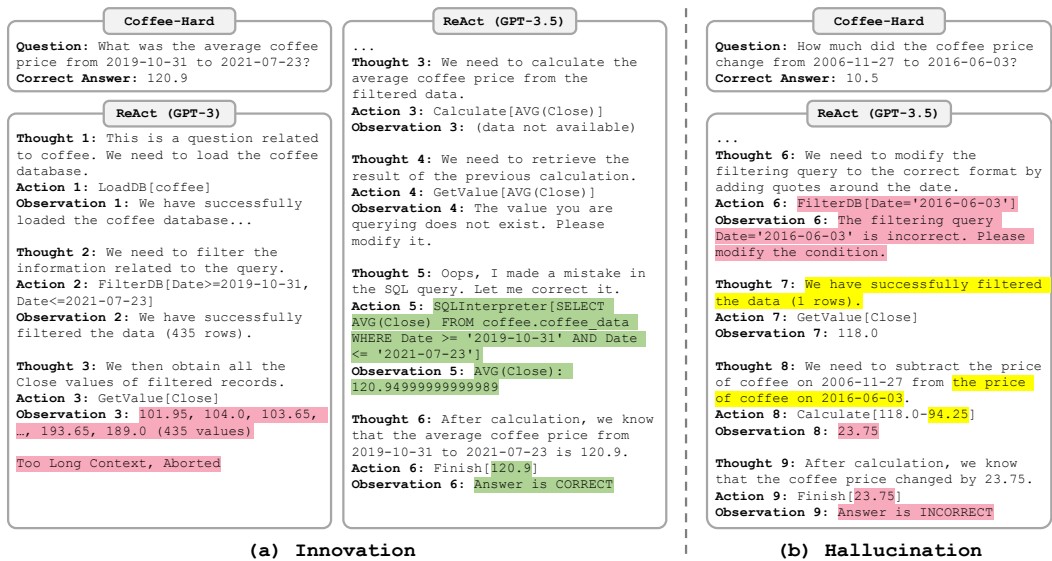

Figure 4: An example of innovation and hallucination when answering hard questions on Coffee data. Actions and observations shrouded in pink are incorrect, whereas those in green are correct. Terms highlighted in yellow signify hallucinations produced by ReAct (GPT-3.5).

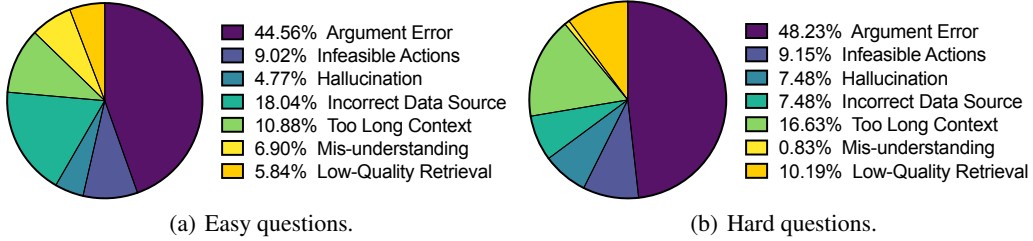

(a) Easy questions.          (b) Hard questions.

Figure 5: Error analysis of ReAct on ToolQA.

- **Infeasible Actions:** The execution of tool calls are infeasible in the environment, often involving new tools that do not exist in the pre-defined tool pool.
- **Too Long Context:** The encoding of interaction history, observations, and tool-use plans exceed the length limitation of GPT family models, resulting in runtime errors;
- **Mis-understanding:** The LLMs cannot understand the observations obtained from external interaction and fail to determine the next steps or generate answers;
- **Low-Quality Retrieval:** This error occurs when the retrieval model fails to extract the relevant information from text corpora, indicating insufficient external knowledge for LLMs to answer questions accurately.

Comparing these error types on easy and hard questions, we find that the overall distribution is similar, though there is a slightly higher rate of hallucination and long-context errors when answering hard questions. This can be attributed to the complexity of hard questions, which often require composing more tools for question answering.

## 5.5 Potential Misuse of ToolQA

We summarize the potential misuse of the ToolQA dataset and Tool-augmented LLMs as follows:

- If LLMs can be trained or prompted to use external tools, they could be prompted to use tools that extract personal information, propagate malware, or provide misleading information;
- There is potential for future LLMs to over-rely on these external tools, sacrificing their intrinsic reasoning abilities. This can make them less versatile in situations where tool use is not feasible;

Table 5: Performance of tool-augmented open-sourced LLM.

| Methods | Coffee-Easy | Coffee-Hard |
|---|---|---|
| LLaMA-2 (13B) | 2.0 | 0.0 |
| ReAct (LLaMA-2, 13B) | 31.0 | 6.2 |
| ReAct (GPT-3.5) | 81.0 | 17.7 |

- As LLMs are prompted to interact with more external systems, the security risks can increase. Malicious actors might find ways to exploit the interactions between LLMs and the external tools they leverage.

## 5.6 Comparing Open-Source LLMs with ChatGPT

Table 5 shows the comparison of vanilla LLaMA-2, tool-augmented LLaMA-2, and tool-augmented ChatGPT. There is a significant performance gap between vanilla LLaMA-2 and tool-augmented LLaMA-2, which is consistent with what we have observed on closed-source LLMs in Section 4.2. In terms of the ability to use external tools for answering the questions, we found that LLaMA-2 is indeed lagging behind ChatGPT. The prompts tailored for tool-augmented LLMs tend to be complicated and lengthy, containing tool descriptions, few-shot examples, and interaction history with the environment. Such long contexts make it difficult for LLaMA-2 to understand complex instructions hidden inside.

## 6 Conclusion and Recommendation

We have developed ToolQA, a dataset that assesses the ability of Large Language Models (LLMs) in using external tools for solving complex problems. ToolQA is curated by an automated three-phase process, including reference data collection, template-based question generation, and programmatic answer generation. This pipeline is general and can be expanded to incorporate external knowledge corpora in different domains. We tested both standard LLMs and tool-augmented LLMs on ToolQA. Our experiments showed that even the strongest model achieved limited performance on the hard questions of ToolQA. Our analysis found that current tool-augmented LLMs tend to make errors such as incorrect tool calls and using incorrect data sources. These issues could be potentially addressed by fine-tuning using a collection of tool-use corpora with open-source LLMs. In the future, we plan to collect high-quality tool-use sequences to fine-tune open-source LLMs, and subsequently evaluating their performance on ToolQA.

It is important to note that the reported performance of closed-source LLMs, such as ChatGPT, is based on specific versions of these models. As these closed-source models undergo further development and updates, the results may change accordingly. We advocate two strategies to mitigate the issue: (1) We plan to continuously include the performance of newly released models in our repository, with the collective efforts of both our team and peer researchers of the community. We intend to integrate results from different versions of such product APIs and document the identifiers of each version (*e.g.*, 0314, 0613, *etc.*) to promote reproducibility. (2) The reported performance of closed-source LLMs should be regarded primarily as a reference, and making comparisons with them is optional. Instead, we encourage the community to use the performance of open-source LLMs as the primary baseline when employing the ToolQA benchmark. This will promote reproducibility and consistency of the evaluation on ToolQA over time.

## Acknowledgments and Disclosure of Funding

This work was supported in part by NSF (IIS2008334, IIS-2106961, CAREER IIS-2144338), ONR (MURI N00014-17-1-2656), IDEaS Cyberinfrastructure Resources, and Microsoft Accelerate Foundation Models Research Program.

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

# A    Additional Related Works

| Methods | Tool Numbers | Tool Categories | # Tool/Task | Reasoning | Instruction Type | Task |
|---------|-------------|-----------------|-------------|-----------|------------------|------|
| *Single-Tool Methods* | | | | | | |
| CoT [65] | 1 | - | 1 | Generation | Prompting | QA |
| Lila [38] | 1 | math/code | 1 | Generation | Prompting | MathQA |
| Program-of-Thought [6] | 1 | code | 1 | Generation | Prompting | TabQA |
| Code4Struct [63] | 1 | code | 1 | Generation | Prompting | Event Extraction |
| PAL [13] | 1 | code | 1 | Generation | Prompting | MathQA |
| MathPrompt [15] | 1 | code | 1 | Generation | Prompting | MathQA |
| ToolFormer [56] | 5 | Basic | 1 | Generation | PR & FT | QA |
| GraphToolFormer [77] | 5 | Graph | 1 | Human Info | PR & FT | Graph |
| Talm [46] | - | Basic | 1 | Generation | PR & FT | QA |
| *Multi-Tool Methods* | | | | | | |
| WebGPT [40] | 10 | Web Operation | >1 | Feedback | Fine-tuning | QA |
| HuggingGPT [57] | >10 | Vision | >1 | Human Info | Prompting | VQA |
| Chameleon [32] | >10 | code, nlp, cv | >1 | Human Info | Prompting | ScienceQA, TabQA |
| GeneGPT [20] | 38 | NCBI APIs | >1 | Generation | Prompting | Gene Tasks |
| ART [45] | 8 | code/math/retriever | >1 | Human Feedback | Prompting | BigBench |
| ReAct [76] | 3 | retriever | >1 | Feedback | PR & FT | QA, AlfWorld, WebShop |
| MM-ReAct [73] | >10 | vision | >1 | Feedback | Prompting | CV tasks |
| Visual ChatGPT [69] | >10 | vision | >1 | Feedback | Prompting | CV tasks |

Table 6: A comparison of methods that leverage LLMs for Tool-use.

We list the state-of-the-art related works in tool-augmented LLMs in Table 6. All of them can be categorized into two groups: (1) single-tool methods, that focus on making a single API call perfect in the solution; (2) multi-tool methods, that emphasize more on studying how to compose different tools together to solve a challenging problem. ToolQA is more suitable for the evaluation of the second category to test the inherent logical reasoning behind different tools. Additionally, there exist other notable contributions [64, 25, 59] within the realm of decision-making that specifically emphasize the planning capabilities of expansive language models. These endeavors can be regarded as methods affiliated with tools, wherein the actions within generated plans are analogous to distinct tools utilized for specific purposes.

# B    Data Sources

## B.1    Different Data Source Introduction

- **Flight Status (2022-2023)**[5] contains almost all flight information of airlines between 2022 and 2023, which is too contemporary for LLMs' internal knowledge.
- **Daily Coffee Price (2000-2022)**[6] contains the daily price of coffee, ranging from 2000 to 2022, where the information is too contemporary and detailed for LLMs' internal knowledge.
- **Yelp Business Data**[7] is a subset of Yelp's business data across 8 metropolitan areas in the USA and Canada, where the information is too detailed for LLMs' internal knowledge.
- **Airbnb Open Data**[8] is a subset of Airbnb activities in New York, where the information is too detailed for LLMs' internal knowledge.
- **DBLP Citation Network (V14)**[9] constructs the graph based on the records after 2020. The author-author and paper-paper relations are formulated as two separate graphs.
- **GSM8k**[10] is a dataset of 8.5K high-quality linguistically diverse grade school math word problems. We sample the questions from the error cases made by ChatGPT on the original dataset to make sure that the questions cannot be easily handled with its internal knowledge.
- **SciREX**[11] is a challenging dataset for document-level information extraction based on a collection of full-length machine-learning scientific papers.

---

[5] https://www.kaggle.com/datasets/robikscube/flight-delay-dataset-20182022?select=Combined_Flights_2022.csv

[6] https://www.kaggle.com/datasets/psycon/daily-coffee-price

[7] https://www.kaggle.com/datasets/yelp-dataset/yelp-dataset?select=yelp_academic_dataset_business.json

[8] https://www.kaggle.com/datasets/arianazmoudeh/airbnbopendata

[9] https://www.aminer.org/citation

[10] https://github.com/openai/grade-school-math

[11] https://github.com/allenai/SciREX

- **Agenda** is our own synthetic dataset to model the real-world personal agenda data. To avoid the privacy issue, we first create names, events, and dates with ChatGPT and then randomly compose them to form 10000 different records. To create a pure-text personal agenda corpus, we feed each of the records into ChatGPT, containing generated agenda for virtual characters. More Details can be seen in Appendix B.2.

### B.2 Generation Details of Agenda Dataset

As mentioned in § 3.2, personal or private data serves as a significant external knowledge source. There exist applications that have been designed with plugins and external tools specifically querying this type of data, such as AI personal assistants on daily agenda. Nevertheless, we recognize that this data often intersects with sensitive areas, and hence, privacy concerns are paramount. To address these issues, we automatically synthesize a personal agenda corpus. This not only ensures that the large language models (LLMs) have not been previously exposed to the data but also eliminates any possibility of them inadvertently memorizing the information within their internal knowledge.

In the synthetically generated personal agenda corpus, each entry follows the pattern: "`NAME` performs `EVENT` at `TIME` on `DATE`", incorporating key elements such as names, events, dates, and time slots. To begin, we employ ChatGPT to virtually generate these elements. More precisely, we create 100 unique names, 10000 distinctive events each associated with corresponding time slots within a day, and span all possible dates from 01/01/2022 through 12/31/2022. Following this, we commence the random assembly of these generated elements to formulate personal agenda entries. For every event-time pair generated, we randomly select from the pool of 100 names and possible dates to construct each record. This process yields a total of 9,494 unique personal agenda entries. To transform this corpus into an accessible external database for model querying, we transcribe each record into a comprehensible natural language description. Prompts designed for agenda data generation are listed in Appendix F.2.

## C   Easy Question Templates

### C.1   Flights

We design the following 10 templates:

- What was the departure time of the {CARRIER}{NUMBER} flight from {ORIGIN} to {DEST} on {ORIGIN}?
- Was the flight {CARRIER}{NUMBER} from {ORIGIN} to {DEST} cancelled on {ORIGIN}?
- What is the flight number of the {AIRLINE} flight from {ORIGIN} to {DEST} on {ORIGIN}?
- How long was the different between the CRS-recorded departure time and actual departure time of the {CARRIER}{NUMBER} flight from {ORIGIN} to {DEST} on {ORIGIN}?
- How long did {CARRIER}{NUMBER} delay when arrival on {DEST}?
- How many extra minutes did the {CARRIER}{NUMBER} flight take from {ORIGIN} to {DEST} on {ORIGIN}?
- What was the local arrival time of the {CARRIER}{NUMBER} flight from {ORIGIN} to {DEST} on {ORIGIN}?
- What was the CRS-recorded arrival time of the {CARRIER}{NUMBER} flight from {ORIGIN} to {DEST} on {ORIGIN}?
- How long was the flight {CARRIER}{NUMBER} from {ORIGIN} to {DEST} on {ORIGIN}?
- How many minutes did the {CARRIER}{NUMBER} flight take to taxi in on {DATE}?

### C.2   Coffee

We design the following 8 templates:

- What was the daily coffee price opening on {DATE}?
- What was the lowest coffee price on {DATE}?
- What was the highest coffee price on {DATE}?
- What was the daily coffee price closing on {DATE}?
- What was the trading volume of coffee on {DATE}?

- What was the percentage change in coffee price on {DATE}, based on the difference between the opening and closing prices?
- Was {DATE} a bearish or bullish day for coffee price?
- What was the range of coffee price on {DATE}, based on the difference between the high and low prices?

## C.3 Yelp

We design the following 11 templates for the Yelp dataset:

- What is the address of {NAME} in the area of postal code {POSTAL-CODE}?
- What city is {NAME} located in {STATE}?
- What state is {NAME} located in?
- What is the postal code of {NAME} in the area with postal code {POSTAL-CODE}, {CITY}, {STATE}?
- What is the star rating of {NAME} in the area with postal code {POSTAL-CODE}, {CITY}, {STATE}?
- How many reviews does {NAME} receive in the area with postal code {POSTAL-CODE}, {CITY}, {STATE}, received?
- Is {NAME} still open in the area with postal code {POSTAL-CODE}, {CITY}, {STATE}?
- Does {NAME} require appointment in the area with postal code {POSTAL-CODE}, {CITY}, {STATE}?
- What are the hours of operation for {NAME} in the area with postal code {POSTAL-CODE}, {CITY}, {STATE}?
- What categories does {NAME} belong to, in the area with postal code {POSTAL-CODE}, {CITY}, {STATE}?
- What are the coordinates of {NAME} in the area with postal code {POSTAL-CODE}, {CITY}, {STATE}?

## C.4 Airbnb

We design the following 10 templates for easy questions on Airbnb dataset:

- What is the host's name for {NAME} in {NEIGHBOURHOOD}?
- How many days are {NAME} (id: {ID}) available during a year (365 days)?
- What is the room type of {NAME} (id: {ID}) in {NEIGHBOURHOOD}?
- What is the price of {NAME} (id: {ID}) in {NEIGHBOURHOOD}?
- What is the minimum number of nights for {NAME} (id: {ID}) in {NEIGHBOURHOOD}?
- When did {NAME} (id: {ID}) in {NEIGHBOURHOOD} constructed?
- How many reviews does {NAME} (id: {ID}) in {NEIGHBOURHOOD} have?
- What is the last review date for {NAME} (id: {ID}) in {NEIGHBOURHOOD}?
- What is the review rate number for {NAME} (id: {ID}) in {NEIGHBOURHOOD}?
- What is the average number of reviews per month for {NAME} (id: {ID}) in {NEIGHBOURHOOD}?

## C.5 SciREX

We design the following 1 templates for easy questions on SciREX dataset:

- What is the corresponding {METRIC} score of the {METHOD} method on {DATASET} dataset for {TASK} task?

## C.6 Agenda

We design the following 5 templates for easy questions on Agenda dataset:

- What did {NAME} do from {START-TIME} to {END-TIME} on {DATE}?
- Where did {EVENT} that {NAME} attended take place on {DATE}?
- When did {NAME} attend {EVENT} on {DATE}?
- How long did {NAME} attend {EVENT} on {DATE}?
- Who attended {EVENT} between {START-TIME} and {END-TIME} on {DATE} in {LOCATION}?

## C.7 DBLP

We design the following 10 templates for easy questions on DBLP dataset:

- Who are the authors of {TITLE}?
- What organization is {AUTHOR} from?
- How many pages is {TITLE}?
- How many papers did {TITLE} cite in the DBLP citation network?
- How many papers did papers in the DBLP citation network cite {TITLE}?
- How many collaborators does {AUTHOR} have in the DBLP citation network?
- How many papers did {AUTHOR} and {AUTHOR} write together in the DBLP citation network?
- What papers did {AUTHOR} write in the DBLP citation network?
- How many papers did {AUTHOR} write in the DBLP citation network?
- What venue did {AUTHOR} and {AUTHOR} collaborate most in the DBLP citation network?

## C.8 GSM8K

The questions are randomly sampled from the ChatGPT errors in GSM8K dataset without following some templates. Thus, we cannot offer any question templates for GSM8K.

# D    Hard Question Templates

## D.1    Flights

- What percentage of the flights from {ORIGIN} were delayed on {FLIGHTDATE}?
- What is the average delay time of all the flights that departed from {ORIGIN} on {FLIGHTDATE}?
- How many flights were diverted on {FLIGHTDATE}?
- How many flights with a distance greater than 500 miles on {FLIGHTDATE}?
- What is the average airtime of the flights from {ORIGIN} to {DEST} host by {AIRLINE}?
- How many flights from {ORIGIN} to {DEST} host by {AIRLINE}?
- What is the average flight time of {CARRIER}{NUMBER}?
- What is the fastest flight from {ORIGIN} to {DEST} on {FLIGHTDATE}?
- What is the average speed of {CARRIER}{NUMBER} from {ORIGIN} to {DEST}?
- What is the total number of flights operated by {AIRLINE} on {FLIGHTDATE}?

## D.2    Coffee

- What was the highest coffee price from {START-DATE} to {END-DATE}?
- What was the lowest coffee price from {START-DATE} to {END-DATE}?
- What was the average coffee price from {START-DATE} to {END-DATE}?
- How much did the coffee price change from {START-DATE} to {END-DATE}?
- What was the percentage change in coffee price on {DATE} compared to the previous day?
- On which date from {START-DATE} to {END-DATE} was the difference between the highest and lowest coffee prices the greatest?
- What was the average daily volume of coffee traded from {START-DATE} to {END-DATE}?
- On which date from {START-DATE} to {END-DATE} did the coffee price have the highest increase compared to the previous day?
- How many times from {START-DATE} to {END-DATE} did the coffee price increase compared to the previous day?
- What was the percentage increase in coffee price from {START-DATE} to {END-DATE}?
- What was the coffee price range from {START-DATE} to {END-DATE}?

### D.3 Yelp

We design the following 10 templates for hard questions in Yelp Dataset.

- How many {CATEGORY} businesses are there in {CITY}, {STATE}?
- How many businesses are there in {POSTALCODE} area of {CITY}, {STATE}?
- Which {CATEGORY} business has the highest star rating in {CITY}, {STATE}?
- Which {CATEGORY} business has the highest review count in {CITY}, {STATE}?"
- What is the average review counts of businesses within a 5-mile radius from {NAME}?
- Which is the nearest {CATEGORY} business to {NAME}?
- Can you recommend a {CATEGORY} business with the highest star rating within a 5-mile radius of {ADDRESS}?
- How many businesses are not open currently in {CITY}?
- What is the average star rating of {CATEGORY} businesses in {CITY}?
- Which region has most bussinesses in {CITY}, {STATE}?

### D.4 Airbnb

We design the following 10 templates for hard questions on Airbnb dataset.

- What is the total price at least if you want to stay at {NAME} in {NEIGHBOURHOOD} for {NUMBER} nights?
- How many airbnbs are there in {NEIGHBOURHOOD}?
- What is the average price of airbnbs in {NEIGHBOURHOOD}?
- What is the average review rates within 5 miles from {NAME} in {NEIGHBOURHOOD}?
- How much proporion of airbnbs in {NEIGHBOURHOOD} have a flexible cancellation policy?
- How much does it cost per night to stay at the most expensive entire home/apt in {NEIGHBOURHOOD}?
- How many airbnbs are there in {NEIGHBOURHOOD} that have a review rate higher than 4?
- Can you recommend me a hotel room with the lowest price in {NEIGHBOURHOOD}?
- Can you recommend me a private room with the highest review rate that can host at least 2 people in {NEIGHBOURHOOD}?
- Can you recommend a shared room with the lowest price within 10 miles from {LONGITUDE} longitude and {LATITUDE} latitude?

### D.5 SciREX

We design the following 4 templates for hard questions on SciREX dataset:

- What is the corresponding {METRIC} score of the {METHOD} method on {DATASET} dataset for {TASK} task?
- On which dataset does the {METHOD} method achieve the highest {METRIC} score for {TASK} task?
- Which method achieves the highest {METRIC} score on {DATASET} dataset for {TASK} task?
- On what metrics is the {METHOD} method evaluated on {DATASET} dataset for {TASK} task?
- Which datasets is {METHOD} method evaluated on for {TASK} task?

### D.6 Agenda

We design the following 5 templates for hard questions on Agenda dataset:

- How many events happen on {DATE} in the agenda table?
- Who is unavailable between {START-TIME} and {END-TIME} on {DATE} in the agenda table?
- When should I schedule a meeting with {NAME} from 9:00 AM to 6:00 PM on {DATE} in the agenda table?
- What events does {NAME} have on {DATE} in the agenda table?
- How many dates in the agenda table have {NAME} scheduled?

### D.7   DBLP

We design the following 10 templates for hard questions on DBLP dataset:

- What keywords does {AUTHOR} focus on most in the DBLP citation network?
- How many people does {AUTHOR-1} need to know at least to know {AUTHOR-2} in the DBLP citation network?
- How many common collaborators does {AUTHOR-1} have with {AUTHOR-2}?
- Which is the most cited paper written by {AUTHOR} in the DBLP citation network?
- Which collaborator does {AUTHOR} have the most citations with in the DBLP citation network?
- Which venue does {AUTHOR} publish the most papers in the DBLP citation network?
- How many accumulated citations do papers collaborated by {AUTHOR-1} and {AUTHOR-2} have in the DBLP citation network?
- How many papers in all do {AUTHOR} and his/her collaborators have in the DBLP citation network?
- Who collaborated with {AUTHOR} most in the DBLP citation network?
- What institutions participated in the study of {TITLE} in the DBLP citation network?

## E   Code Examples of Programmatic Answer Generation

Below is an example of programmatic answer generation. The example code is answering the question of "What percentage of the flights from {ORIGIN} were delayed on {FLIGHTDATE}?". More details of the programmatic answers can be seen in the public code.

```python
def solution(data, flightdate, origin):
    num_total =len(data.loc[(data["FlightDate"] ==flightdate) & (data["Origin"] ==
                                            origin)])
    num_cancelled =len(data.loc[(new_data["FlightDate"] ==flightdate) &
                                       (data["Origin"] ==origin) &
                                       (data["Cancelled"] ==True)])
    if num_cancelled >0:
        question ="What percentage of the flights from {} were delayed on
                                          {}?".format(origin, flightdate)
        answer ="{:.1f}".format(num_cancelled /num_total *100)+"%"
```

## F   Additional Implementation Details

### F.1   Implementation Details

All experiments are conducted on *CPU*: Intel(R) Core(TM) i7-5930K CPU @ 3.50GHz and *GPU*: NVIDIA GeForce RTX A5000 GPUs using python 3.8, Huggingface 4.6.0 and Pytorch 1.10. We keep the parameter $\text{top\_p} = 1.0$ and temperature $t = 1.0$ for calling ChatGPT APIs [43] for the question generation part.

### F.2   Prompts

#### F.2.1   Prompts for Agenda Data Generation

The prompts used for virtual name generation:

```
<Agenda_Name_Gen> Prompt
You are an AI assistant to answer questions.
Can you list 100 English Names?
```

The prompts used for virtual events generation:

________________ <Agenda_Events _Gen> Prompt ________________

```
You are an AI assistant for text generation.
Generate 100 detailed agenda events, including the event, start time, end time, and
location. Please make the events as diverse as possible and make sure these events
can happen in real life. Make sure the location is a detailed name that may exist
in real life. Make sure the dates are selected from 2022/01/01 to 2023/01/01.

Example:
Doctor's appointment - 9:00 AM - 11:00 AM - ABC Medical Center
Yoga class - 10:30 AM - 11:30 AM - Yoga Studio Downtown

Generate 100 more detailed agendas that do not conflict with the previous ones.
```

The prompts used to convert the agenda records into natural language descriptions:

________________ <Agenda_Gen> Prompt ________________

```
Please use natural language to describe the event in the agenda with the following
information:

Name: NAME
Date: DATE
Event: EVENT
Start Time: START-TIME
End Time: END-TIME
Location: LOCATION
```

### F.2.2 Prompts for Methods

The prompts used in ReAct [76]:

________________ <ReAct> Prompt ________________

```
Question: How many extra minutes did the DL1575 flight take from ATL to MCO
    on 2022-01-12?
Thought 1: This is a question related to flights. We need to load the flights
    database.
Action 1: LoadDB[flights]
Observation 1: We have successfully loaded the flights database, including the
    following columns: FlightDate, Airline, Origin, Dest, Cancelled, Diverted,
    CRSDepTime, DepTime, DepDelayMinutes, DepDelay, ArrTime, ArrDelayMinutes,
    AirTime, CRSElapsedTime, ActualElapsedTime, Distance, Year, Quarter, Month,
    DayOfWeek, Marketing_Airline_Network, Operated_or_Branded_Code_Share_Partners,
    DayofMonth, DOT_ID_Marketing_Airline, IATA_Code_Marketing_Airline,
    Flight_Number_Marketing_Airline, Operating_Airline, DOT_ID_Operating_Airline,
    IATA_Code_Operating_Airline, Tail_Number, Flight_Number_Operating_Airline,
    OriginAirportID, OriginAirportSeqID, OriginCityMarketID, OriginCityName,
    OriginState, OriginStateFips, OriginStateName, OriginWac, DestAirportID,
    DestAirportSeqID, DestCityMarketID, DestCityName, DestState, DestStateFips,
    DestStateName, DestWac, DepDel15, DepartureDelayGroups, DepTimeBlk, TaxiOut,
    WheelsOff, WheelsOn, TaxiIn, CRSArrTime, ArrDelay, ArrDel15, ArrivalDelayGroups,
    ArrTimeBlk, DistanceGroup, DivAirportLandings.
Thought 2: We need to filter the information related to the query.
Action 2: FilterDB[Flight_Number_Marketing_Airline=1575, FlightDate=2022-01-12,
    Origin=ATL, Dest=MCO]
Observation 2: We have successfully filtered the data (1 row).
Thought 3: We then need to know the departure delayed time.
Action 3: GetValue[DepDelay]
Observation 3: -7.0
Thought 4: We then need to know the arrival delayed time.
Action 4: GetValue[ArrDelay]
Observation 4: -17.0
Thought 5: To compute the extra minutes, we need to subtract the departure delayed
    time from the arrival delayed time.
Action 5: Calculate[(-17)-(-7)]
Observation 5: -10
```

```
Thought 6: After calculation, we know that the answer is -10.
Action 6: Finish[-10]

Question: Michael had 58 golf balls. On tuesday, he lost 23 golf balls. On
    wednesday, he lost 2 more. How many golf balls did he have at the end of
    wednesday?
Thought 1: Maybe I can write a python program to solve this problem.
Action 1: PythonInterpreter[# solution in Python:
def solution():
    # Michael had 58 golf balls. On tuesday, he lost 23 golf balls. On wednesday,
    # he lost 2 more. How many golf balls did he have at the end of wednesday?
    golf_balls_initial = 58
    golf_balls_lost_tuesday = 23
    golf_balls_lost_wednesday = 2
    golf_balls_left = golf_balls_initial - golf_balls_lost_tuesday - \
        golf_balls_lost_wednesday
    result = golf_balls_left
    return result]
Observation 1: 33
Thought 2: After execution of Python Code, we know that the answer is 33.
Action 2: Finish[33]

Question: What is the corresponding Mean_IoU score of the FRRN method on Cityscapes
    dataset for Semantic_Segmentation task?
Thought 1: The question is asking some performance information about Mean_IoU score
    of the FRRN method on Cityscapes dataset for Semantic_Segmentation task, we
    need to retrieve some useful information from the scirex database.
Action 1: RetrieveScirex[Mean_IoU score of the FRRN method on Cityscapes dataset
    for Semantic_Segmentation task]
Observation 1: The other stream undergoes a sequence of pooling operations to obtain
    robust features for recognition . The two streams are coupled at the full image
    resolution using residuals . Without additional processing steps and without
    pretraining , our approach achieves an intersection - over - union score of
    71.8 % on the Cityscapes dataset .
Thought 2: As the approach achieves an intersection - over - union score of 71.8 %
    on the Cityscapes dataset, the answer should be 71.8%.
Action 2: Finish[71.8%]

Question: When was the paper Learning the Principle of Least Action with
    Reinforcement Learning. published?
Thought 1: The question is asking some basic information of a dblp paper. We need
    to load the DBLP graph.
Action 1: LoadGraph[dblp]
Observation 1: DBLP graph is loaded.
Thought 2: The question is asking the published date of a paper, we need to check
    the node from the PaperNet in DBLP graph.
Action 2: NodeCheck[PaperNet, Learning the Principle of Least Action with
    Reinforcement Learning.]
Observation 2: {'title': 'Learning the Principle of Least Action with Reinforcement
    Learning.', 'year': 2021, 'venue': {'raw': 'AAAI Spring Symposium - MLPS'},
    'n_citation': 0, 'keywords': [], 'doc_type': 'Conference', 'page_start': '',
    'page_end': ''}
Thought 3: The published date of the paper is 2021.
Action 3: Finish[2021]

Question: How many collaborators does Chao Zhang have in the DBLP graph?
Thought 1: The question is asking the number of collaborators of a person, we
    need load the DBLP graph.
Action 1: LoadGraph[dblp]
Observation 1: DBLP graph is loaded.
Thought 2: The question is asking the collaborators of a person, we need to
    check the neighboring nodes from the AuthorNet in DBLP graph.
Action 2: NeighbourCheck[AuthorNet, Chao Zhang]
Observation 2: ['YUHUI YUAN', 'Rao Fu', 'Lang Huang', 'Weihong Lin', 'X Chen',
    'Jingdong Wang']
```

```
Thought 3: The number of collaborators of Chao Zhang is 6.
Action 3: Finish[6]

Question: How many papers does Chao Zhang and Weihong Lin have in common in the
    DBLP graph?
Thought 1: The question is asking the number of common papers of two persons, we
    need load the DBLP graph.
Action 1: LoadGraph[dblp]
Observation 1: DBLP graph is loaded.
Thought 2: The question is asking the common papers of two persons, we need to
    check the edges between them from the PaperNet in DBLP graph.
Action 2: EdgeCheck[PaperNet, Chao Zhang, Weihong Lin]
Observation 2: {'weight': 1, 'papers': ['HRFormer: High-Resolution Vision
    Transformer for Dense Predict.'], 'n_citation': [95]}
Thought 3: The number of common papers of Chao Zhang and Weihong Lin is 1.

Question: Where did Stephen's Opera performance take place?
Thought 1: The question is asking the location of Stephen's Opera performance
    from agenda.
Action 1: RetrieveAgenda[Stephen's Opera performance]
Observation 1: On January 29, 2022, there will be an opera performance at the Lyric
    Opera House, featuring Stephen. The show will start at 7:00 PM and end at
    9:00 PM. It promises to be a wonderful evening of beautiful music and powerful
    performances in a stunning venue. Come and experience the magic of opera at its
    finest!
Thought 2: The event happened in Lyric Opera.
Action 2: Finish[Lyric Opera]

Question: What was the trading volume of coffee on 2000-01-14?
Thought 1: Maybe I can write a SQL query to solve this problem.
Action 1: SQLInterpreter(SELECT Volume FROM coffee.coffee_data WHERE Date =
    '2000-01-14';
Observation 1: Volume: 10115
Thought 2: The volume of coffee on 2000-01-14 is 10115.
Action 2: Finish[10115]
```

The prompts used in Chameleon [32]:

```
─────────────────────────────┤ <Chameleon> Prompt ├──────────────────────────────
You need to act as a policy model, that given a question and a modular set,
    determines the sequence of modules that can be executed sequentially can solve
    the question.

The modules are defined as follows:

- Calculate[formula]: This module calculates a given formula and returns the
    result. It takes in a mathematical formula and returns the calculated result.
    Normally, we only consider using "Calculate" when the question involves
    mathematical computations.

- RetrieveAgenda[keyword]: This module retrieves an agenda related to a specific
    keyword and returns it. It takes in a keyword and returns the corresponding
    agenda. Normally, we only consider using "RetrieveAgenda" when the question is
    about specific actions or tasks related to a topic.

- RetrieveScirex[keyword]: This module retrieves paragraphs from machine learning
    papers related to the specified keyword and returns them. It takes in a keyword
    and returns the relevant paragraphs. Normally, we only consider using
    "RetrieveScirex" when the question involves understanding specific concepts
    in machine learning.

- LoadDB[DBName]: This module loads a database specified by the database name and
    returns the loaded database. It takes in a database name and returns the
    corresponding database. The DBName can be one of the following: flights/
    coffee/airbnb/yelp. Normally, we only consider using "LoadDB" when the
```

question requires data from a specific structured dataset.

- FilterDB[column_name, relation, value]: This module filters a database by a specified column name, relation, and value, and then returns the filtered database. It takes in a column name, a relation, and a value, and returns the filtered database. Normally, we only consider using "FilterDB" when the question requires a specific subset of data from a structured dataset.

- GetValue[column_name]: This module returns the value of a specified column in a database. It takes in a column name and returns its value. Normally, we only consider using "GetValue" when the question requires a specific piece of data from a structured dataset.

- LoadGraph[GraphName]: This module loads a graph specified by the graph name and returns the loaded graph. It takes in a graph name and returns the corresponding graph. Normally, we only consider using "LoadGraph" when the question involves understanding or navigating specific graph structures.

- NeighbourCheck[GraphName, Node]: This module lists the neighbors of a specified node in a graph and returns the neighbors. It takes in a graph name and a node, and returns the node's neighbors. Normally, we only consider using "NeighbourCheck" when the question involves understanding relationships in a graph structure.

- NodeCheck[GraphName, Node]: This module returns the detailed attribute information of a specified node in a graph. It takes in a graph name and a node, and returns the node's attributes. Normally, we only consider using "NodeCheck" when the question requires information about a specific entity in a graph.

- EdgeCheck[GraphName, Node1, Node2]: This module returns the detailed attribute information of the edge between two specified nodes in a graph. It takes in a graph name and two nodes, and returns the attributes of the edge between them. Normally, we only consider using "EdgeCheck" when the question involves understanding the relationship between two entities in a graph.

- SQLInterpreter[SQL]: This module interprets a SQL query and returns the result. It takes in a SQL query and returns the result of the query. Normally, we only consider using "SQLInterpreter" when the question requires data manipulation and extraction from a structured dataset.

- PythonInterpreter[Python]: This module interprets Python code and returns the result. It takes in Python code and returns the result of the code execution. Normally, we only consider using "PythonInterpreter" when the question requires complex computations or custom data manipulation.

- Finish[answer]: This module returns the final answer and finishes the task. This module is the final module in the sequence that encapsulates the result of all previous modules.

Below are some examples that map the problem to the modules.

Question: How many extra minutes did the DL1575 flight take from ATL to MCO on 2022-01-12?

Modules: ["LoadDB[flights]", "FilterDB[Flight_Number_Marketing_Airline=1575, FlightDate=2022-01-12, Origin=ATL, Dest=MCO]", "GetValue[DepDelay]", "GetValue[ArrDelay]", "Calculate[(-17)-(-7)]", "Finish[-10]"]

Question: Michael had 58 golf balls. On tuesday, he lost 23 golf balls. On wednesday, he lost 2 more. How many golf balls did he have at the end of wednesday?

Modules: ["PythonInterpreter[# solution in Python:\n\ndef solution():\n # Michael had 58 golf balls. On tuesday, he lost 23 golf balls. On wednesday, he lost 2 more. How many golf balls did he have at the end of wednesday?\n

```
       golf_balls_initial = 58\n golf_balls_lost_tuesday = 23\n
       golf_balls_lost_wednesday = 2\n golf_balls_left =
       golf_balls_initial - golf_balls_lost_tuesday - golf_balls_lost_wednesday\n
       result = golf_balls_left\n return result]", "Finish[33]"]

Question: What is the corresponding Mean_IoU score of the FRRN method on Cityscapes
     dataset for Semantic_Segmentation task?

Modules: ["ScirexRetrieve[Mean_IoU score of the FRRN method on Cityscapes dataset
     for Semantic_Segmentation task]", "Finish[71.8%]"]

Question: When was the paper Learning the Principle of Least Action with
     Reinforcement Learning. published?

Modules: ["LoadGraph[dblp]", "NodeCheck[PaperNet, Learning the Principle of
     Least Action with Reinforcement Learning.]", "Finish[2021]"]

Question: How many collaborators does Chao Zhang have in the DBLP graph?

Modules: ["LoadGraph[dblp]", "NeighbourCheck[AuthorNet, Chao Zhang]", "Finish[6]"]

Question: How many papers does Chao Zhang and Weihong Lin have in common in
     the DBLP graph?

Modules: ["LoadGraph[dblp]", "EdgeCheck[PaperNet, Chao Zhang, Weihong Lin]",
     "Finish[1]"]

Question: Where did Stephen's Opera performance take place?

Modules: ["AgendaRetrieve[Stephen's Opera performance]", "Finish[Lyric Opera]"]

Question: What was the trading volume of coffee on 2000-01-14?

Modules: ["SQLInterpreter[SELECT Volume FROM coffee.coffee_data WHERE Date =
     '2000-01-14']", "Finish[10115]"]

Now, you need to act as a policy model, that given a question and a modular set,
     determines the sequence of modules that can be executed sequentially can
     solve the question.
```

# G  Key Information of ToolQA

## G.1  Dataset Documentations

The dataset is provided in *jsonl* format. Each task corresponds to two files: easy and hard (*e.g.*, "flight-easy.jsonl" and "flight-hard.jsonl", *etc.*). Each data point contains the following fields:

- qid: the unique identifier for the question-answer pair;
- question: the question to query;
- answer: the corresponding ground-truth answer to question.

## G.2  Intended Uses

ToolQA is intended for researchers in machine learning and related fields to innovate novel methods for tool-augmented large language models (LLMs). We also aim to help developers to test their plugins on our dataset.

## G.3  Hosting and Maintenance Plan

ToolQA codebase is hosted and version-tracked via GitHub. It will be permanently available under the link https://github.com/night-chen/ToolQA. The download link of all the datasets can be found in the GitHub repository.

ToolQA is a community-driven and open-source initiative. We are committed and have resources to maintain and actively develop ToolQA in the future. We plan to grow ToolQA to include more tasks, tools, and more baseline methods. We welcome external contributors.

## G.4 Licensing

We license our work using Apache 2.0[12]. All the datasets will be publicly released through the aforementioned GitHub link.

## G.5 Statement

The authors will bear all responsibility in case of violation of rights.

---

[12]https://www.apache.org/licenses/LICENSE-2.0

