# OpenReview forum: "ToolQA: A Dataset for LLM Question Answering with External Tools"
_NeurIPS.cc/2023/Track/Datasets_and_Benchmarks — NeurIPS 2023 Datasets and Benchmarks Poster_

### Official Review · Reviewer_3WCN · 2023-07-19
**Interesting discussion, but need more solid evaluation pipelines and support & evaluation for open-sourced models**

**Rating:** 6
**Confidence:** 4

**Strengths:**

- This study is well-motivated and addresses a few critical problems in the existing LLMs, considering a risk that the external data used for evaluation may have already been exposed to the models during the pre-training.
- To discuss how well LLMs can use external tools to provide correct answers, the authors designed experiment pipeline such that LLMs would not correctly answer the questions if the models do not use the right tool and right data sources.
- The result analysis is very detailed and provides interesting findings, especially how a method failed to correctly answer questions (incorrect tool calls, incorrect data sources, too long context, etc).

**Additional Feedback:**

## Checklist
- Most of the answers are just [Yes, No, or N/A] and should be followed by justifications/comments.
- The reviewer could not find the training details in Appendix G. It should be clarified.
> Did you specify all the training details (e.g., data splits, hyperparameters, how they were chosen)? [Yes] See in Appendix G.

## Questions
- What is the definition of the difficulty of a question (lines 180)? Is it quantified somehow or completely subjective?
- How did the authors find data from public sources that LLMs do not memorize (lines 113 and 114)?
- Some of the code in the repository were edited after the submission deadline. Do the changes affect the reproducibility of the work?
- Is there any persistent dereferenceable identifier e.g., DOI provided? It is highly recommended as mentioned in the CfP

**Clarity:**

- At lines 132 and 133, how did the authors confirm that ChatGPT cannot answer questions from GSM8K correctly with its own mathematical reasoning ability?
- "However, both strategies" at line 162 -> "However, both the strategies"
- "Table 3 and 4" at lines 221 and 232 should be "Tables 3 and 4"
- "interested in include" at line 314 should be "interested in including"

**Correctness:**

This paper repeatedly mentions that the authors are attempting to minimize the overlap between the benchmark data and LLMs' pre-training data, and the reviewer finds it very important and appreciates it. However, the reviewer did not find any quantitative discussion regarding the overlap in this study. At the same time, since all the base models used in this study seem like black-box product APIs and details of the training data are unclear, it sounds impossible to discuss such an overlap. How did the authors attempt to minimize the overlap?

For generating questions, if the authors use either GPT-3 or GPT-3.5 for "Human-guided Question Generation with LLMs" at lines 50 and 51, it may give the LLMs more favor at inference time. But, it seems that the section does not explain what LLMs were used to generate questions, and the reviewer could not assess if the dataset is constructed in a sound way.

**Documentation:**

- The reviewer has a concern in the reproducibility of this work as the GitHub repository does not provide instructions to perform the experiment in an end-to-end manner (data preparation -> inference -> evaluation -> result analysis). On top of providing the complete code (including Chameleon code), the authors should offer clear instructions.
- The dataset documentation in the repository seems very limited. At least some basic stats (e.g., Table 1) should be added to README.md.
- The following author statement must be included, according to [the CfP](https://neurips.cc/Conferences/2023/CallForDatasetsBenchmarks), but the reviewer could not find the statement.
> Author statement that they bear all responsibility in case of violation of rights, etc., and confirmation of the data license.

**Limitations:**

The reviewer did not find that the authors have adequately addressed the limitations and potential negative societal impact of their work.

The most critical limitation of this work the reviewer found is that all the baselines considered in this study are based on black-box product APIs. While different approaches are considered, using only the same black-box models makes the reviewer feel how scientifically meaningful the resulting findings are.

Also, the code repository still misses Chameleon code.

**Opportunities For Improvement:**

- For the evaluation part, it should be formalized in either the paper of supplementary material, how the authors normalize both the ground-truth answers and the model predictions. Since the success rates are based on the exact match between the normalized answers, this process should be very important for the evaluation process. But, the reviewer did not find any descriptions about the normalization process.
- All the tested methods seem based on black-box product APIs whose training data or model architectures are not known. This work should consider at least 1 or 2 open-sourced LLMs for baselines. This should also be important to help the community build on the authors' work, instead of imposing the use of Open AI service (not for free) on the community.

**Relation To Prior Work:**

Sections 2.1 and 2.2 clearly discuss the differences from the previous contribution.

**Summary And Contributions:**

This paper introduces a new dataset called ToolQA, which is designed to evaluate large language models (LLMs)' ability to leverage external tools for answering questions given that existing LLMs face critical issues such as hallucination and weak numerical reasoning. The dataset covers six different contexts: Temporal, Spatial, Mathematical, Social, Scientific, and Personal. Experimental results with four different methods (LLMs and tool-augmented LLMs) for ToolQA indicate that ReAct (GPT-3) and ReAct (GPT-3.5) performed the best in terms of success rate on easy and hard questions, respectively.
The result analysis also show that the existing tool-augmented LLMs are not good at making correct tool calls and correct data source.

---

> ### Author Response · Authors · 2023-08-21
> **Rebuttal by Authors - PART I**
>
> **[Improvement 1] For the evaluation part, it should be formalized in either the paper of supplementary material, how the authors normalize both the ground-truth answers and the model predictions. Since the success rates are based on the exact match between the normalized answers, this process should be very important for the evaluation process. But, the reviewer did not find any descriptions about the normalization process.**
>
> Thank you for the insightful comments. We normalized the predictions and ground-truth answers using the following set of rules:
> - We normalized different time string formats, (e.g, converting “18:06” and “1806.0” to “1806”.);
> - For price-related questions, we normalize the units by removing price units (e.g., USD, $);
> - We removed all the punctuations from both the model predictions and ground-truth answers;
> - We normalized the article usage (e.g., a, an, the) by removing all articles from both the model predictions and ground-truth answers;
> - We normalized the white spaces by trimming multiple spaces into a single space;
>
> As most of the predictions and answers are numerical values or entities, these normalization rules address most of the false negative cases during matching. Furthermore, we have manually examined through all the matching results to ensure the faithfulness of the reported model performance. We have added more clarifications about the normalization part in Section 4.1 of the updated version.
>
> **[Improvement 2] All the tested methods seem based on black-box product APIs whose training data or model architectures are not known. This work should consider at least 1 or 2 open-sourced LLMs for baselines. This should also be important to help the community build on the authors' work, instead of imposing the use of Open AI service (not for free) on the community.**
>
> Thanks for the comment. We have conducted new experiments on the newly-released open-source models Falcon (40B) and LLaMA-2 (13B, 70B).  The results are as follows:
> | Easy Questions | Flight | Coffee | Agenda | Yelp | DBLP | SciREX | GSM8K | Airbnb | Average |
> |----------------|:------:|:------:|:------:|:----:|:----:|:------:|:-----:|:------:|:-------:|
> | LLaMA-2 (13B)  |   0.0  |   2.0  |   0.0  |  5.0 |  1.0 |   0.0  |  9.0  |   1.0  |   2.3   |
> | Falcon (40B)  |  1.0  | 1.0  |  2.0  | 8.0 | 1.0 |  0.0  | 8.0  |  5.0  |   3.3  |
> | LLaMA-2 (70B)  |  2.0    |   6.0  |   5.0  |   15.0  | 0.0 |   0.0  |  9.0   |   4.0 | 5.1 |
>
> | Hard Questions  | Flight | Coffee | Agenda | Yelp | DBLP | SciREX | Airbnb | Average |
> |-----------------|:------:|:------:|:------:|:----:|:----:|:------:|:------:|:-------:|
> | LLaMA-2 (13B)   |   1.0  |   0.0  |   0.0  |  4.0 |  1.0 |   5.0  |   1.0  |   1.7   |
> | Falcon (40B)  | 1.0   | 0.0  |  0.0  | 4.0 | 1.0 | 6.0  | 1.0   |  1.9   |
> | LLaMA-2 (70B)   |   1.0  |   0.0  |   0.0  |  4.0 |  1.0 |   4.0  |   3.0  |   1.9   |
>
> Based on the results, the open-sourced LLMs exhibit limited reasoning capabilities across all tasks and struggle with questions in ToolQA. This is because they lack external tool-use ability to access external data. Furthermore, open-sourced LLMs still show worse performance than ChatGPT due to the limitation of fewer parameters. We also observed that open-sourced LLMs often struggle to comprehend and follow instructions strictly. As a result, integrating LLaMA-2 with baselines like ReAct shows significantly lower performance compared with ReAct (GPT-3.5) and may need further improvements. We have added our new results and discussions in Section 4.2 of the updated version.
>
> **[Limitation 1] The most critical limitation of this work the reviewer found is that all the baselines considered in this study are based on black-box product APIs. While different approaches are considered, using only the same black-box models makes the reviewer feel how scientifically meaningful the resulting findings are.**
>
> Thanks for your valuable suggestion. Please refer to our response to [Improvement 2]for more details.
>
> **[Limitation 2] Also, the code repository still misses Chameleon code.**
>
> Thanks for pointing this out. We have updated our code repository to include the source code of Chameleon.

---

> > ### Author Response · Authors · 2023-08-21
> > **Rebuttal by Authors - PART II**
> >
> > **[Correctness 1] This paper repeatedly mentions that the authors are attempting to minimize the overlap between the benchmark data and LLMs' pre-training data, and the reviewer finds it very important and appreciates it. However, the reviewer did not find any quantitative discussion regarding the overlap in this study. At the same time, since all the base models used in this study seem like black-box product APIs and details of the training data are unclear, it sounds impossible to discuss such an overlap. How did the authors attempt to minimize the overlap?**
> >
> > The baseline performance of vanilla ChatGPT provides quantifiable evidence of minimal overlap between its training data and ToolQA's external data. When feeding ToolQA's questions directly to ChatGPT, the success rates reveal its ability to use internal knowledge for external data queries. The notably low performance of vanilla ChatGPT on most tasks suggests a minimal overlap between the pre-training data of ChatGPT and the external data of ToolQA. We have added more clarification in Section 4.2 of the updated version.
> >
> > On the other hand, we attempted to reduce the overlap between the LLMs' pre-training data and benchmark data via careful data source selection. As highlighted in Lines 143-153 (updated version):
> > - The flight, coffee, and DBLP data are after September 2021, which lies beyond the training timelines of ChatGPT and GPT-4.
> > - The Yelp and Airbnb datasets consist of non-textual corpora, enriched with spatial details like geographical coordinates—elements potentially absent from ChatGPT's training data.
> > - The agenda data is a synthetic dataset, which contains virtual personal schedule information. LLMs would not have encountered such unique information during training.
> >
> > In summary, our selected data sources cover recent information beyond the training timeframe, non-textual specifics potentially missing from training datasets, and synthetic details not publicly accessible, thereby excluding them from the LLMs' training data.
> >
> > **[Correctness 2] For generating questions, if the authors use either GPT-3 or GPT-3.5 for "Human-guided Question Generation with LLMs" at lines 50 and 51, it may give the LLMs more favor at inference time. But, it seems that the section does not explain what LLMs were used to generate questions, and the reviewer could not assess if the dataset is constructed in a sound way.**
> >
> > As indicated in Line 183 (updated version), we employed ChatGPT (gpt-3.5-turbo) to generate question templates. To avoid introducing bias or favor during the question generation, we prompted ChatGPT to generate “diverse and challenging template questions”. With the specified prompt, we generated around 50 template questions for each external data source (e.g., flight, coffee, etc.). To ensure the soundness of these questions, we examined through and excluded any templates that fulfill the following conditions (flight data used for examples):
> > - Templates that ChatGPT might answer using its inherent knowledge base (e.g., “What is the distance between LAX and SFO?”)
> > - Templates posing questions about information not present in the external data (e.g., “What is the average price from LAX to SFO?”).
> >
> > Employing such rigorous criteria, we handpicked around 10 top-tier templates for every data source and difficulty tier. Furthermore, the low success rate of vanilla ChatGPT also proves there is no favor or bias introduced into inference during the question generation process. We have added more clarification in Section 3.3 of the updated version.
> >
> > **[Clarity 1] At lines 132 and 133, how did the authors confirm that ChatGPT cannot answer questions from GSM8K correctly with its own mathematical reasoning ability?**
> >
> > We input all the GSM8K questions into ChatGPT five times, utilizing majority voting to determine the predictions for each question. Subsequently, we chose 100 questions where the predictions diverged from ChatGPT's responses, constituting the GSM8K subset within ToolQA.
> >
> > **[Clarity 2,3,4] Typos.**
> >
> > Thank you for pointing these out. We will fix the typos in the updated version.
> >
> > **[Documentation 1] The reviewer has a concern in the reproducibility of this work as the GitHub repository does not provide instructions to perform the experiment in an end-to-end manner (data preparation -> inference -> evaluation -> result analysis). On top of providing the complete code (including the Chameleon code), the authors should offer clear instructions.**
> >
> > We have added clearer instructions on how to run the experiments in the updated version of the GitHub repository (https://github.com/night-chen/ToolQA/blob/main/benchmark/README.md).
> >
> > **[Documentation 2] The dataset documentation in the repository seems very limited. At least some basic stats (e.g., Table 1) should be added to README.md.**
> >
> > We have added more statistics from the paper to the repository, including the dataset statistics in Table 1 and the main results tables.

---

> > > ### Author Response · Authors · 2023-08-21
> > > **Rebuttal by Authors - PART III**
> > >
> > > **[Documentation 3] The following author statement must be included, according to the CfP, but the reviewer could not find the statement. Author statement that they bear all responsibility in case of violation of rights, etc., and confirmation of the data license.**
> > >
> > > Thank you for the reminder. We have added the statement in Appendix G.5 of the updated version.
> > >
> > > **[Checklist 1] Most of the answers are just [Yes, No, or N/A] and should be followed by justifications/comments.**
> > >
> > > We have modified the answers to the checklist section in the updated version to include more justifications and comments.
> > >
> > > **[Checklist 2] The reviewer could not find the training details in Appendix G. It should be clarified. Did you specify all the training details (e.g., data splits, hyperparameters, how they were chosen)? [Yes] See in Appendix G.**
> > >
> > > Thank you for pointing this out. Sorry, this is a typo. We offer the implementation details in Appendix F.1. As our dataset and models do not involve a training process, we only report the hardware details and the hyperparameters of calling OpenAI APIs.
> > >
> > > **[Questions 1] What is the definition of the difficulty of a question (lines 180)? Is it quantified somehow or completely subjective?**
> > >
> > > As detailed in lines 202-207 (updated version), the distinction between easy and hard questions is based on the quantity of information they probe. Easy questions target singular pieces of information—be it a row in a table, a segment of text in a corpus, or a node in a graph. In contrast, hard questions require reasoning across several information fragments, such as multiple rows, nodes, or passages. Logically it is more challenging to piece together multiple data elements.
> > >
> > > **[Questions 2] How did the authors find data from public sources that LLMs do not memorize (lines 113 and 114)?**
> > >
> > > Thank you for your insightful question. Please refer to our response to [Correctness 1] for more details.
> > >
> > > **[Questions 3] Some of the code in the repository were edited after the submission deadline. Do the changes affect the reproducibility of the work?**
> > >
> > > The changes are mainly about data cleaning and privacy information removal (e.g., API keys), which will not influence the reproducibility of the work. Moreover, we are committed to regularly updating the repository, ensuring that ToolQA remains a valuable resource for fellow researchers in the community.
> > >
> > > **[Questions 4] Is there any persistent dereferenceable identifier e.g., DOI provided? It is highly recommended as mentioned in the CfP.**
> > >
> > > Thank you for the kind reminder. As is specified in CfP, we either use persistent dereferenceable identifier for dataset or a code repository for code. We chose to use the code repository of github (https://github.com/night-chen/ToolQA) to maintain both code and data.

---

> > > > ### Comment · Reviewer_3WCN · 2023-08-23
> > > >
> > > > First of all, the reviewer thanks the authors for their detailed responses.
> > > >
> > > > The reviewer is almost ready for finalizing the review and leaves a few more comments, responding to some of the authors' comments.
> > > >
> > > > > As indicated in Line 183 (updated version), we employed ChatGPT (gpt-3.5-turbo) to generate question templates. To avoid introducing bias or favor during the question generation, we prompted ChatGPT to generate “diverse and challenging template questions”.
> > > >
> > > > This is a good example to highlight the reviewer's concern about dependencies on black-box product APIs.
> > > >
> > > > There is no guarantee that works for ChatGPT (as a question generator). Not limited to ChatGPT, using a specific prompt or instruction to address a problem for a task (generating questions in this case) cannot be a justification or negate that it may give the LLMs more favor at inference time, unless the actual models are trained in that manner. Either ChatGPT or other closed product APIs do not have information about the performance for the specific task, while the APIs' behaviors change overtime without (public) version controls and cannot reproduce the behavior.
> > > >
> > > > > Based on the results, the open-sourced LLMs exhibit limited reasoning capabilities across all tasks and struggle with questions in ToolQA. This is because they lack external tool-use ability to access external data. Furthermore, open-sourced LLMs still show worse performance than ChatGPT due to the limitation of fewer parameters. We also observed that open-sourced LLMs often struggle to comprehend and follow instructions strictly. As a result, integrating LLaMA-2 with baselines like ReAct shows significantly lower performance compared with ReAct (GPT-3.5) and may need further improvements. We have added our new results and discussions in Section 4.2 of the updated version.
> > > >
> > > > The reviewer recognizes and appreciates extra effort that the authors made for the requested experiments during the discussion period.
> > > > As baselines, the open-sourced models seem more appropriate for benchmarking since those results should be more reproducible and transparent than the black-box APIs, thus the reviewer do not necessarily see open-sourced models underperforming the black-box APIs in a negative way.
> > > > Also, how did the authors conclude from the results that "This is because they lack external tool-use ability to access external data."? i.e., how did the authors identify lack of external tool-use ability as the cause of the lower performance?
> > > > For instance, it is fairly possible that different prompts may work better for some open-source models.

---

> > > > > ### Author Response · Authors · 2023-08-25
> > > > > **Author Response to Follow-up Comments**
> > > > >
> > > > > **[Favor at Inference Time] There is no guarantee that works for ChatGPT (as a question generator). Not limited to ChatGPT, using a specific prompt or instruction to address a problem for a task (generating questions in this case) cannot be a justification or negate that it may give the LLMs more favor at inference time, unless the actual models are trained in that manner.**
> > > > >
> > > > > Thank you for your further comments. We are unsure about the reviewer's viewpoint regarding the concept of "favor at inference time." It appears that the new comment:
> > > > > >"using a specific prompt cannot be a justification that it may give the LLMs more favor at inference time,"
> > > > >
> > > > > contradicts the reviewer's previous statement:
> > > > > >"it may give the LLMs more favor at inference time."
> > > > >
> > > > > Although all the questions are generated by ChatGPT, they have been verified by us as natural-style questions, which need to be answered with external data. Most LLMs can comprehend the questions, but the key to answering them lies in leveraging external knowledge. In other words, it is the tool-use ability that matters, instead of the style. The fact that ChatGPT is used for question generation does not necessarily give it an advantage. This can be reflected by the low success rate of vanilla ChatGPT in our experiments.
> > > > >
> > > > > **[Behavior Changes of Closed-Source LLMs] Either ChatGPT or other closed product APIs do not have information about the performance for the specific task, while the APIs' behaviors change overtime without (public) version controls and cannot reproduce the behavior.**
> > > > >
> > > > > Thanks for your comment. The change-of-behavior issue is not a key concern for two reasons. First, our main contribution is the development of the challenging benchmark ToolQA to the community. In the future, we expect to continuously update the performance of more recent models on this benchmark, with the collective efforts of both our team and peer researchers in the community. Second, our experiment is mainly to show that mainstay LLMs have difficulties in solving the challenging questions in ToolQA. This can be reflected by the limited performance of GPT models, as well as our new results on LLaMA-2.
> > > > >
> > > > > **[External Tool-Use Ability] Also, how did the authors conclude from the results that "This is because they lack external tool-use ability to access external data."? i.e., how did the authors identify lack of external tool-use ability as the cause of the lower performance? For instance, it is fairly possible that different prompts may work better for some open-source models.**
> > > > >
> > > > > Thank you for your questions. We would like to clarify that the limited performance of vanilla LLMs, both open- and closed-source, stems from their lack of external information. From the main results in Tables 3 and 4, there are significant performance gaps between vanilla LLMs and their tool-augmented counterparts. We also observe the same gap in our new experiments on LLaMA-2 (13B):
> > > > > | Methods | Coffee-easy | Coffee-Hard |
> > > > > |:-----------|:-----------------:|:-----------------:|
> > > > > | LLaMA-2 (13B) | 2.0 | 0.0|
> > > > > | ReAct (LLaMA-2, 13B) | 31.0 | 6.0 |
> > > > > | ReAct (GPT-3.5) | 81.0 | 17.7 |
> > > > >
> > > > > In terms of the ability to use external tools for answering the questions, we found that LLaMA-2 is indeed lagging behind ChatGPT from the above results. We observed that LLaMA-2 faces challenges in understanding complex instructions hidden in long contexts. The prompts tailored for tool-augmented LLMs tend to be complicated and lengthy, containing tool descriptions, few-shot examples, and interaction history with the environment. Due to the above issues, the significant performance gap and the observed errors are unlikely to be resolved simply through prompt engineering.

---

> > > > > > ### Comment · Reviewer_3WCN · 2023-08-27
> > > > > >
> > > > > > What the reviewer meant by "using a specific prompt cannot be a justification that it may give the LLMs more favor at inference time" is that the following author response does not explain why the questions generated by ChatGPT can be diverse and challenging as we do not know ChatGPT is suitable for the task. But, the authors justify the generated questions by explaining that they prompted ChatGPT to do so, which does not sound like convincing to the reviewer.
> > > > > >
> > > > > > >  To avoid introducing bias or favor during the question generation, we prompted ChatGPT to generate “diverse and challenging template questions”
> > > > > >
> > > > > > ---
> > > > > >
> > > > > > > In the future, we expect to continuously update the performance of more recent models on this benchmark, with the collective efforts of both our team and peer researchers in the community.
> > > > > >
> > > > > > This is actually a reason that the reviewer opposes using closed APIs as baselines. If we know the behaviors of such APIs may change and we cannot reproduce the results, we should not use them baselines (or should add an important note to the paper (e.g., conclusion) that such closed APIs should not be used as strict baselines as those may not be reproducible). Imagine this work is recognized in the community, and subsequent studies want to build on this work, but the baseline results may change over time. Researchers would not want to refer to such results as baselines if they know that the results may change over time.
> > > > > >
> > > > > > On the other hands, it was good to learn from the previous authors' response that they confirmed a similar trend using open-sourced models, regarding external tool-use ability. The reviewer will improve the score if the revision and camera-ready include the new results AND explicitly emphasize in the main body (e.g., conclusion) that communities should refer to open-sourced models' performance as proper baselines and comparisons with closed APIs are optional.

---

> > > > > > > ### Author Response · Authors · 2023-08-28
> > > > > > > **Paper Draft Updated**
> > > > > > >
> > > > > > > Thank you for your additional feedback. We have revised the manuscript according to your suggestions. The changes (highlighted in blue) include:
> > > > > > > - Added clarification about question template generation and manual selection in **Section 3.3** and the normalization process in **Section 4.1**
> > > > > > > - Added new results about open-source LLMs and result analysis in **Tables 3, 4**, **Section 4.2**, and **Appendix I**
> > > > > > > - Updated the conclusion in **Section 6** by highlighting the importance of using open-source LLMs as baselines
> > > > > > >
> > > > > > > We hope that these revisions and our previous responses have addressed your concerns.

---

> > > > > > > > ### Comment · Reviewer_3WCN · 2023-08-29
> > > > > > > >
> > > > > > > > The reviewer appreciates the authors' effort at the last minute.
> > > > > > > > Given that the manuscript and repository were both updated, the reviewer increased the rating by 2.

---

### Official Review · Reviewer_HHeV · 2023-07-21

**Rating:** 9
**Confidence:** 3
**Correctness:** Claims are sound and correct
**Clarity:** The paper is clearly written.

**Strengths:**

The paper studies an important research problem: the evaluation of the tool-use ability of LLMs.

ToolQA creation process is scalable and automated, the 13 specialized tools are diverse and comprehensive.

Evaluation results of LLMs on the ToolQA dataset show the proposed benchmark is effective. The proposed ToolQA benchmark is challenging for state-of-the-art LLMs and can faithfully evaluate the LLMs' ability to use external tools for question answering.


**Additional Feedback:**

Discussion of limitations is preferred

**Documentation:**

ToolQA is clearly documented

**Limitations:**

Any discussion of potential misuse of the ToolQA dataest? Like LLMs can be fooled/prompted to use malicious tools.

**Opportunities For Improvement:**

Since the ToolQA creation process is scalable and automated, it would be better to include even more diverse tasks to compressively evaluate LLMs since LLMs are evolving and their abilities are becoming stronger.



**Relation To Prior Work:**

The paper clearly discusses tool-use benchmarks and datsets

**Summary And Contributions:**

The paper introduces a dataset called ToolQA to benchmark the ability to use external tools for large language models (LLMs). The authors build ToolQA via a scalable and automated process to create 13 specialized tools spanning text, database, math, graph, code, and systems. The ToolQA requires external knowledge to answer questions and is designed to measure and diagnose the tool-use reasoning capabilities of LLMs. The authors perform an extensive evaluation and analysis of state-of-the-art open-source LLMs and commercial LLMs like ChatGPT. The results show ChatGPT and CoT prompting has low success rates while tool-enhanced LLMs like ReAct and Chmeleon are better if equipped with external tools, both of which indicate the ToolQA benchmark remains challenging.

---

> ### Author Response · Authors · 2023-08-21
> **Rebuttal by Authors**
>
> **[Improvement 1] Since the ToolQA creation process is scalable and automated, it would be better to include even more diverse tasks to compressively evaluate LLMs since LLMs are evolving and their abilities are becoming stronger.**
>
> Thank you so much for your valuable suggestion. We recognize the significance of incorporating a wider variety of tasks to evaluate the progression of LLMs. And we should note that ToolQA is also a continually evolving dataset, and we are committed to its consistent enhancement and expansion. At present, we believe that the hard questions pose a substantial challenge for tool-augmented LLMs. Moving forward, we plan to delve into a broader array of tasks, drawing from diverse external knowledge sources and integrating additional functional tools to enrich the dataset.
>
> **[Limitations] Any discussion of potential misuse of the ToolQA dataset? Like LLMs can be fooled/prompted to use malicious tools.**
>
> We summarize the potential misuse of the ToolQA dataset and Tool-augmented LLMs as follows:
> - If LLMs can be trained or prompted to use external tools, they could be prompted to use tools that extract personal information, propagate malware, or provide misleading information.
> - There's a potential for future LLMs to over-rely on these external tools, sacrificing their intrinsic reasoning abilities. This can make them less versatile in situations where tool use is not feasible.
> - As LLMs are prompted to interact with more external systems, the security risks can increase. Malicious actors might find ways to exploit the interactions between LLMs and the external tools they leverage.
>
> We have added more discussion of potential misuse of the ToolQA dataset in Appendix G.6 of the updated version.

---

### Official Review · Reviewer_MPGH · 2023-07-21
**Dataset for external tool-based question answering**

**Rating:** 7
**Confidence:** 3

**Strengths:**

The dataset excels in several major aspects as below:

- Large-scale dataset: ToolQA's development involves a three-phase automated dataset curation process, ensuring efficiency and minimizing the need for extensive human labeling efforts. The benchmark data in ToolQA is meticulously crafted to have minimal overlap with LLMs' pre-training data, ensuring a more accurate evaluation of LLMs' abilities. ToolQA comprises data from 8 different domains.
- Evaluation: The paper addresses the existing challenge in evaluation methods, which fail to distinguish between LLMs' internal knowledge and their genuine use of external tools for problem-solving. ToolQA includes 13 specialized tools that allow interaction with external knowledge, providing a comprehensive evaluation of LLMs' reasoning abilities when using these tools. Also, the benchmark data in ToolQA is meticulously crafted to have minimal overlap with LLMs' pre-training data, ensuring a more accurate evaluation of LLMs' abilities.
- Insightful Performance Analysis: Experiments on ToolQA reveal the substantial performance gap between standard LLMs and tool-augmented LLMs, highlighting the dataset's effectiveness as a challenging benchmark.

**Additional Feedback:**

N/A

**Clarity:**

The paper is exemplarily well-crafted. The authors skillfully emphasize ToolQA's distinctive strengths, including its robust generation process, rigorous evaluation, and insightful findings, positioning it as a valuable asset for external tool-based question answering.

**Correctness:**

The paper is sound and correct. The evaluation metrics are fair and appropriate.

**Documentation:**

The documentation is clear, complete and easy to follow.

**Limitations:**

Limited Tool Diversity: The current ToolQA dataset includes 13 specialized tools, which may not fully represent the vast array of potential external knowledge sources and tools that exist in real-world applications.
Less-scalable answer generation: It seems that the answers have to be carefully given so that the tools can be used properly, which may involve non-trivial amount of human work, leading to less scalability. Also, the current programmatic answer generation process assumes that each question has a fixed and pre-determined answer, which may not fully capture the complexities of real-world question answering scenarios with ambiguous or context-dependent answers.

**Opportunities For Improvement:**

Expansion of Tool Categories: The current ToolQA dataset includes 13 specialized tools; future improvements could involve expanding the variety of tools to encompass a broader range of external knowledge sources and enhance the dataset's diversity.

Complex Reasoning Challenges: Introducing more complex and nuanced questions in ToolQA would encourage the development of LLMs with improved reasoning abilities, pushing the boundaries of their tool-use capabilities.

Real-World Data Incorporation: Integrating real-world, up-to-date external data sources into ToolQA could make the evaluation more practical and relevant to real-life applications, further challenging LLMs' abilities to interact with constantly changing information.

Multi-Modal Inputs: Incorporating multi-modal inputs, such as images, audio, and videos, in ToolQA could reflect real-world scenarios and stimulate the development of LLMs that can utilize various modalities for question answering.

**Relation To Prior Work:**

Prior works are clearly discussed in this paper.

**Summary And Contributions:**

The paper introduces ToolQA, a dataset created to assess the capacity of Large Language Models (LLMs) in utilizing external tools for question answering. ToolQA includes 13 tools designed to interact with external knowledge for answering questions.

---

> ### Author Response · Authors · 2023-08-21
> **Rebuttal by Authors**
>
> **[Improvement 1] Expansion of Tool Categories: The current ToolQA dataset includes 13 specialized tools; future improvements could involve expanding the variety of tools to encompass a broader range of external knowledge sources and enhance the dataset's diversity.**
>
> Thank you for your insightful comments. Our existing suite of 13 specialized tools already encompasses a broad spectrum of general tools, adept at tackling real-world challenges and synergizing with each other to address complex tasks. However, we recognize the potential to expand this dataset. We will continually upgrade the ToolQA dataset, delving into possible enhancements that span additional real-world scenarios, diverse external data sources, and more tools.
>
> **[Improvement 2] Complex Reasoning Challenges: Introducing more complex and nuanced questions in ToolQA would encourage the development of LLMs with improved reasoning abilities, pushing the boundaries of their tool-use capabilities.**
>
> Thank you for the insightful suggestion. We have incorporated complex and nuanced queries into the 'hard questions' section of ToolQA.  Each of these challenging questions requires the synergy of multiple tools and spans several records in the external data. Our experimental findings indicate that even the most advanced tool-augmented LLMs struggle with these questions, delivering only modest performance.
>
> **[Improvement 3] Real-World Data Incorporation: Integrating real-world, up-to-date external data sources into ToolQA could make the evaluation more practical and relevant to real-life applications, further challenging LLMs' abilities to interact with constantly changing information.**
>
> Thank you for the insightful suggestion. We agree that integrating real-world and up-to-date external data sources into the dataset is important. Currently, the data sources are selected to contain information after 2021, which does not overlap with the training data of ChatGPT. More real-world and up-to-date external data sources, like news, will be explored further in our future efforts.
>
> **[Improvement 4] Multi-Modal Inputs: Incorporating multi-modal inputs, such as images, audio, and videos, in ToolQA could reflect real-world scenarios and stimulate the development of LLMs that can utilize various modalities for question answering.**
>
> Thanks for the valuable comment. The multi-modal inputs, involving images, audio, and videos, are out of the scope of this dataset. ToolQA is tailored to assess the external tool-use capabilities, which can be achieved without delving into multi-modal inputs. That said, we recognize the significance and potential of exploring this avenue in the future.
>
> **[Limitations1] Limited Tool Diversity: The current ToolQA dataset includes 13 specialized tools, which may not fully represent the vast array of potential external knowledge sources and tools that exist in real-world applications.**
>
> Thank you for pointing out the limitations. Please refer to our response to [Improvement 1] for more details.
>
> **[Limitations 2] Less-scalable answer generation: It seems that the answers have to be carefully given so that the tools can be used properly, which may involve a non-trivial amount of human work, leading to less scalability. Also, the current programmatic answer generation process assumes that each question has a fixed and pre-determined answer, which may not fully capture the complexities of real-world question answering scenarios with ambiguous or context-dependent answers.**
>
> The exact answers to the questions are generated automatically through the programmatic tool chains, making the question-answer pairs easy to scale up. To faithfully evaluate the tool-use abilities of tool-augmented LLMs, it is important to have a fixed answer to estimate the performance of the models. Incorporating open-ended questions would make it challenging to determine the accuracy of model predictions. The questions with ambiguous questions are out of the scope of ToolQA.

---

### Official Review · Reviewer_H5AJ · 2023-07-23
**About a dataset ToolQA which is designed to faithfully evaluate LLMs' ability to use external tools for question answering**

**Rating:** 6
**Confidence:** 4
**Correctness:** We commonly regard text-davinci-003 a…
**Clarity:** Yes

**Strengths:**

This paper explores LLMs' ability of external tool using more fairly and explicitly.

**Additional Feedback:**

None

**Documentation:**

Yes

**Opportunities For Improvement:**

This paper proposes a dataset for LLMs' external tool using evaluation. What confuses me is the meaning of "external tool use". It seems that when talking about the problem solving capabilities of LLMs, there are two aspects: memorizing and reasoning. In this paper, the "external tool using" means "utilizing external tools for problem-solving" instead of "simply recalling pre-trained information", which is quite unclear. For the dataset construction, the main issue is that the diversity of questions is poor: 1) the questions are not natural language questions in real-world applications; 2) the templates of questions are limited, which makes the contribution of this paper limited.

**Relation To Prior Work:**

Yes

**Summary And Contributions:**

This paper contributes a dataset ToolQA that is designed to evaluate LLMs' ability of external tool using more faithfully, in other words, to detect if LLMs are good at problem-solving with tools or just using their memorized information. They provide a pipeline of generating such dataset with LLMs, heuristic rules and human labor. They also conduct in-depth analyses of different models and error types.

---

> ### Author Response · Authors · 2023-08-21
> **Rebuttal by Authors**
>
> **[Improvement 1] This paper proposes a dataset for LLMs' external tool using evaluation. What confuses me is the meaning of "external tool use". It seems that when talking about the problem solving capabilities of LLMs, there are two aspects: memorizing and reasoning. In this paper, the "external tool using" means "utilizing external tools for problem-solving" instead of "simply recalling pre-trained information", which is quite unclear.**
>
> Thank you for the valuable comment. The concept of "LLMs’ external tool use" refers to the ability of LLMs to access external APIs for information not present within their model weights (internal knowledge from pre-train data), such as up-to-date data, code execution capability, and proprietary sources. Depending on the nature of these external tools, their usage can augment both memorizing and reasoning capacities of the model. For instance, if the external tool is a retriever drawing relevant data from a vectorbase, it enhances the model's memorization ability. On the other hand, if the tool serves as a code interpreter, it enhances the model's reasoning capabilities. We have also added more clarifications on high-level relations among tool use, reasoning, and memorizing in Section 2.1 of the updated version.
>
> **[Improvement 2] For the dataset construction, the main issue is that the diversity of questions is poor: 1) the questions are not natural language questions in real-world applications; 2) the templates of questions are limited, which makes the contribution of this paper limited.**
>
> We respectfully disagree with the statements in this comment.
> - **“Not natural language questions in real-world applications”**: The data we use are actually curated from real-world data, including coffee prices, flight information, business information, etc. And the question templates are carefully generated via collaboration between human efforts and LLMs. We use prompts like: “*Please list questions that users may have when checking the external data.*”  to ensure our questions align with practical applications. As an illustration of this real-world relevance, we have included a question from the Yelp-hard dataset in ToolQA: “*Can you recommend an Art Supplies business with the highest star rating within a 5-mile radius of 615 Channelside Dr?*”
> - **“The templates are limited”**: Employing templates to craft questions is a common approach in the development of numerous existing question-answering datasets [1,2,3]. The main purpose of involving templates is to guarantee the correctness of the QA pairs. To keep the generalization of our dataset, we prompted ChatGPT to generate around 50 template questions for each external data source. After examining all the templates, we selected 117 most representative and diverse question templates for the entire ToolQA dataset. Thus, with the collaboration between human guidance and ChatGPT, the entire process can be easily applied to generate new question templates on existing or new external data sources, making the templates of questions not limited.
>
> **Reference:**
>
> [1] Zhang, Michael, and Eunsol Choi. "SituatedQA: Incorporating Extra-Linguistic Contexts into QA." Proceedings of the 2021 Conference on Empirical Methods in Natural Language Processing. 2021.
>
> [2] Clark, Peter, Oyvind Tafjord, and Kyle Richardson. "Transformers as soft reasoners over language." Proceedings of the Twenty-Ninth International Conference on International Joint Conferences on Artificial Intelligence. 2021.
>
> [3] Dhingra, Bhuwan, et al. "Time-Aware Language Models as Temporal Knowledge Bases." Transactions of the Association for Computational Linguistics 10 (2022): 257-273.

---

> > ### Comment · Reviewer_H5AJ · 2023-08-29
> > **Thanks for the rebuttal**
> >
> > I would like to thank the authors for their response. The explanation solves my confusion. I have adjusted the rating.
> > For concerns about template questions, I understand the benefits of employing templates and new templates can be generated. The main concern is checking the templates generated by LLMs may be time-consuming and human-extensive. 117 templates may be not enough to check LLMs' tool using ability comprehensively, and extension of the dataset is desired.

---

> ### Author Response · Authors · 2023-08-28
> **Official Comments by Authors**
>
> Dear Reviewer, as the rebuttal phase is ending soon, we wanted to bring to your attention that we have made updates to both the manuscript and the code repository, in addition to our previous responses. We kindly request you to review these updates and provide any further comments you may have.

---

### Official Review · Reviewer_sogC · 2023-07-26
**ToolQA: A Dataset for LLM Question Answering with External Tools**

**Rating:** 7
**Confidence:** 4
**Correctness:** Yes, some more data verification stud…
**Clarity:** Yes

**Strengths:**

- focus on a timely problem to distinguish between questions that can be answered using LLMs’ internal knowledge and those that require external information through tool use.
- introduce a new dataset called ToolQA to evaluate LLMs’ ability to use external tools for question answering.
- semi-automated process for dataset curation, along with several specialized tools designed for interaction with external knowledge in order to answer questions.
- diagnosis of existing tool-use LLMs to highlight their strengths, weaknesses, and potential improvements.
- data publicly available.

**Additional Feedback:**

NA

**Documentation:**

Yes

**Ethics:**

Authors may want to add more clarification on the license of the references they collect in Step 1 of their data creation process.

I am not sure if a separate ethics review is needed.

**Limitations:**

Yes, the error analysis is good.
More on data quality studies and use of instruction paradigm to mitigate errors will be useful.

**Opportunities For Improvement:**

- could not find much details about data verification and  data quality studies. Each step in the data creation process can have errors, specifically using models. Not sure how authors are verifying quality and making sure dataset represents the task they intend to solve?
- 2nd part of the method involve template which can have generalization issues. Not sure author have tried something more free form, say may be by using LLMs.
- It is bit surprising that model performance is so low on these benchmarks. The argument error could probably be fixed by using natural language instruction (https://aclanthology.org/2022.acl-long.244.pdf) along with tool usage, wondering any thoughts on that?

**Relation To Prior Work:**

Many of the instruction-tuning, prompting, decomposition, tool use literature is missing and needs to be added.

E.g. A couple examples on decomposition that are relevant:
https://openreview.net/pdf?id=_nGgzQjzaRy
https://aclanthology.org/2022.emnlp-main.302/
https://arxiv.org/abs/2210.03350

Similarly, there are several other works in instruction-tuning and prompting which needs to be discussed.

**Summary And Contributions:**

This paper addresses a limitation in the current evaluation methods as they do not distinguish between questions that can be answered using LLMs’ internal knowledge and those that require external information through tool use.

- The authors introduce a new dataset called ToolQA, which is designed to faithfully evaluate LLMs’ ability to use external tools for question answering.
- They investigate tool-use LLMs highlighting their strengths, weaknesses, and various potential improvements

---

> ### Author Response · Authors · 2023-08-21
> **Rebuttal by Authors - PART I**
>
> **[Improvement 1] Could not find much details about data verification and data quality studies. Each step in the data creation process can have errors, specifically using models. Not sure how authors are verifying quality and making sure dataset represents the task they intend to solve?**
>
> Thank you for your thoughtful comment. Our data generation process involved human efforts to ensure the quality of the data. When generating the question templates, we went through all templates and eliminated those that fulfill the following conditions (using flight data as examples):
> - Templates that ChatGPT might answer using its inherent knowledge (e.g., “What is the distance between LAX and SFO?”)
> - Templates posing questions about information not present in the external data (e.g., “What is the average price from LAX to SFO?”).
>
> Similarly, when generating answers programmatically, we manually checked the correctness of all the programs. After producing the question-answer pairs, we went through the entire dataset multiple times—typically 3-4 rounds by different coauthors independently—to guarantee the validity of each pair. We have added more clarifications about data verification quality in Section 3.3 of the updated version.
>
> **[Improvement 2] 2nd part of the method involve template which can have generalization issues. Not sure author have tried something more free form, say may be by using LLMs.**
>
> Thanks for this insightful question. Designing templates to generate questions is widely used during the creation of many existing question-answering datasets [1,2,3].  The main purpose of involving templates is to guarantee the correctness of the QA pairs. To keep the generalization of our dataset, we used ChatGPT with human guidance to generate the templates and integrated data from eight diverse sources, covering temporal, spatial, social, and domain-specific information.
>
> We have tried using LLMs to automatically generate all the questions instead of the templates. However, it is quite difficult to guarantee question quality. For example, the LLMs may generate questions on flight data like “What is the average price of the flight from LAX to ATL?”, which is querying nonexistent information in the reference external data.
>
> **[Improvement 3] It is bit surprising that model performance is so low on these benchmarks. The argument error could probably be fixed by using natural language instruction (https://aclanthology.org/2022.acl-long.244.pdf) along with tool usage, wondering any thoughts on that?**
>
> Thank you for the useful suggestion. The emphasis of this paper is on the dataset curation and the assessment of external tool-use capabilities in both vanilla LLMs and mainstay tool-augmented LLMs. While we recognize that numerous techniques can boost the performance of existing methods, our study does not focus on the advancement of these techniques. We encourage further explorations and novel designs on these modules and invite researchers to evaluate them using our dataset in future endeavors.
>
> **[Relation to Prior Work] Many of the instruction-tuning, prompting, decomposition, tool use literature is missing and needs to be added. E.g. A couple examples on decomposition that are relevant: https://openreview.net/pdf?id=_nGgzQjzaRy https://aclanthology.org/2022.emnlp-main.302/ https://arxiv.org/abs/2210.03350 Similarly, there are several other works in instruction-tuning and prompting which needs to be discussed.**
>
> Thanks for pointing out these relevant works. We will add them in the updated version and discuss them. In addition, [5] advocates for breaking down intricate tasks into simpler, optimized prompts, offering insights for multi-step interactions in datasets like ToolQA. [6] explores the potential of tailoring questions to a model’s strengths through human-involved decomposition. Meanwhile, [7] delves into LLMs' challenges with multi-hop reasoning, introducing structured prompting techniques like self-ask. In summary, these decomposition methods have the potential to enhance the planning ability of LLMs, leading to potential improved performance on ToolQA.

---

> > ### Author Response · Authors · 2023-08-21
> > **Rebuttal by Authors - PART II**
> >
> > **References:**
> >
> > [1] Zhang, Michael, and Eunsol Choi. "SituatedQA: Incorporating Extra-Linguistic Contexts into QA." Proceedings of the 2021 Conference on Empirical Methods in Natural Language Processing. 2021.
> >
> > [2] Clark, Peter, Oyvind Tafjord, and Kyle Richardson. "Transformers as soft reasoners over language." Proceedings of the Twenty-Ninth International Conference on International Joint Conferences on Artificial Intelligence. 2021.
> >
> > [3] Dhingra, Bhuwan, et al. "Time-Aware Language Models as Temporal Knowledge Bases." Transactions of the Association for Computational Linguistics 10 (2022): 257-273.
> >
> > [4] Shinn, Noah, Beck Labash, and Ashwin Gopinath. "Reflexion: an autonomous agent with dynamic memory and self-reflection." arXiv preprint arXiv:2303.11366 (2023).
> >
> > [5] Khot, Tushar, et al. "Decomposed Prompting: A Modular Approach for Solving Complex Tasks." The Eleventh International Conference on Learning Representations. 2022.
> >
> > [6] Patel, Pruthvi, et al. "Is a Question Decomposition Unit All We Need?." Proceedings of the 2022 Conference on Empirical Methods in Natural Language Processing. 2022.
> >
> > [7] Press, Ofir, et al. "Measuring and narrowing the compositionality gap in language models." arXiv preprint arXiv:2210.03350 (2022).

---

### Author Response · Authors · 2023-08-28
**General Response and Summarize of Updates to Manuscripts**

We appreciate the reviewers’ comments that we focus on an important and timely problem (reviewers sogC, HHeV) with a new challenging dataset (reviewers sogC, MPGH, H5AJ), detailed experiments on state-of-the-art models (reviewers MPGH, HHeV, 3WCN), and in-depth analysis of the results given by tool-augmented LLMs (reviewers sogC, MPGH, 3WCN). We would like to echo such acknowledgments and summarize the key contributions of the paper:
- **Creation of a new challenging dataset:** Our curated ToolQA is a new dataset that faithfully evaluates the LLMs’ ability to use external tools for question answering, presenting challenges to state-of-the-art LLMs.
- **Semi-automated generation process:** We proposed a human-LLM collaboration process for question generation to achieve both high-quality and scalable generation.
- **Benchmarking state-of-the-art methods on ToolQA:** We have compared the performance of various state-of-the-art vanilla and tool-augmented LLMs, covering both closed- and open-source ones.
- **Thorough diagnosis of tool-augmented LLMs:** We have conducted an in-depth diagnosis of existing tool-use LLMs to highlight their strengths, weaknesses, and potential improvements.

To address some concerns from the reviewers, we have uploaded a revised manuscript according to the feedback. The revisions include:
- Added more related works about planning and memorizing abilities of LLMs in Section 2.1 (**Reviewers sogC and H5AJ**);
- Added clarification about the generation and manual selection of the question in Section 3.3 (**Reviewers sogC, H5AJ, 3WCN**);
- Added new results about open-source LLMs and result analysis in Tables 3, 4, Section 4.2 , and Appendix I (**Reviewer 3WCN**);
- Updated the conclusion in Section 6 by highlighting the importance of using open-source LLMs as baselines (**Reviewers MPGH, HHeV, 3WCN**);
- Added discussion on potential misuse of tool-augmented LLMs in Appendix H  (**Reviewer HHeV**).

---

### Decision · Program_Chairs · 2023-09-22

**Decision:**

Accept (Poster)

**Comment:**

ToolQA is a dataset for evaluating tools in question answering.  This will be a timely addition to the Benchmarks track and will help push forward research in tool use, an area which is rapidly growing.